# Symmetry Discovery Beyond Affine Transformations

**Ben Shaw**
Utah State University
Logan, UT 84322
*ben.shaw@usu.edu*

**Abram Magner**
University at Albany
Albany, NY 12222
*amagner@albany.edu*

**Kevin R. Moon**
Utah State University
Logan, UT 84322
*kevin.moon@usu.edu*

## Abstract

Symmetry detection can improve various machine learning tasks. In the context of continuous symmetry detection, current state of the art experiments are limited to detecting affine transformations. Under the manifold assumption, we outline a framework for discovering continuous symmetry in data beyond the affine transformation group. We also provide a similar framework for discovering discrete symmetry. We experimentally compare our method to an existing method known as LieGAN and show that our method is competitive at detecting affine symmetries for large sample sizes and superior than LieGAN for small sample sizes. We also show our method is able to detect continuous symmetries beyond the affine group and is generally more computationally efficient than LieGAN.

## 1 Introduction

Approaching various machine learning tasks with prior knowledge, commonly in the form of symmetry present in datasets and tasks, has been shown to improve performance and/or computational efficiency [1, 2, 3, 4]. While research in symmetry detection in data has enjoyed recent success [5], the current state of the art leaves room for improvement, particularly with respect to the detection of continuous symmetry [6]. Herein, we seek to address two issues at hand: primarily, the inability of current methods to detect continuous symmetry beyond the affine group; secondly, the apparent increase of difficulty in identifying continuous symmetries when compared with discrete symmetries.

Our methodology is summarized as follows. First, given a dataset, certain quantities of interest, hereafter referred to as *machine learning functions*, are identified, such as an underlying probability distribution which generates the dataset, a classification or regression function, or a dimensionality reduction function. Second, we estimate tangent vector fields which annihilate the machine learning functions. The estimated vector fields themselves are generators of continuous symmetries. Thus estimating the vector fields provides an indirect estimate of the continuous symmetries. Third, we use the estimated vector fields to estimate functions which, in addition to the machine learning functions, are annihilated by the vector fields. The discovered functions define a feature space which is invariant with respect to the continuous symmetries implied by the vector field generators. On this constructed feature space, the outcome of any machine learning task is, by definition, invariant with respect to the symmetries generated by the estimated vector fields, and we show in examples how symmetry-based feature engineering can improve performance on machine learning tasks.

Vector fields have been used to describe symmetry in the context of machine learning before [7]. However, our use of vector fields differs from previous approaches for two reasons. First, we fully exploit the vector fields as objects which operate on machine learning functions directly. Second, we estimate vector fields associated with continuous symmetries beyond affine transformations. **Our main contributions are the following**: 1) We show that our method can detect symmetry in a number of cases previously examined with other methods, with our method offering a number of computational advantages. 2) We also show that our method can be used to detect continuous symmetries beyond the affine group, experimenting with both synthetic and real data.

38th Conference on Neural Information Processing Systems (NeurIPS 2024).

## 2   Related Work and Background

Some background is necessary for our work, beginning with a general description of symmetry. Throughout, we assume a dataset $\mathcal{D} = \{x_1, \ldots, x_r\}$ with $x_i \in \mathbb{R}^n$. Various machine learning functions of interest may exist for a given dataset, such as an underlying probability distribution or class probabilities. Generally speaking, the object of symmetry detection in data is to discover functions $S : \mathbb{R}^n \to \mathbb{R}^n$ that preserve a machine learning function of interest. That is, for a machine learning function $f$, $f \circ S = f$. We consider a continuous symmetry of a particular machine learning function to be a transformation $S$ which is continuously parameterized, such as a rotation in a plane by an angle $\theta$, with $\theta$ being the continuous parameter. We consider a discrete symmetry of a machine learning function to be a transformation which is not continuously parameterized, such as a planar rotation by a fixed angle, or a reflection of a point in the plane about a straight line. We primarily deal with continuous symmetry, though discrete symmetry detection is discussed in Appendix D.3.

Comparatively early work on symmetry detection in machine learning seems to have focused primarily on detecting symmetry in image and video data [8, 9], where symmetries described by straight-line and rotational transformations were discovered. Other work has made strides in symmetry discovery by restricting the types of symmetries being sought. In one case, detection was limited to compact Abelian Lie groups [10]. Another case uses meta-learning to discover any *finite* symmetry group [6]. Finite groups have also been used in symmetry discovery in representation learning [11].

Other work has focused on detecting affine transformation symmetries and encoding the discovered symmetries automatically into a model architecture. Three such methods identify Lie algebra generators to describe the symmetries, as we do here. For example, *augerino* [12] attempts to learn a distribution over augmentations, subsequently training a model with augmented data. The *Lie algebra convolutional network* [7], which generalizes Convolutional Neural Networks in the presence of affine symmetries, uses infinitesimal generators represented as vector fields to describe the symmetry. SymmetryGAN [13] has also been used to detect rotational symmetry [5].

*LieGAN* appears to represent the current state of the art for continuous symmetry detection. LieGAN is a generative-adversarial network intended to return infinitesimal generators of the continuous symmetry group of a given dataset [5]. LieGAN has been shown to detect continuous affine symmetries, including even transformations from the Lorentz group. It has also been shown to identify discrete symmetries such as rotations by a fixed angle.

None of these related works have attempted to detect continuous symmetries beyond affine transformations. Continuous symmetry detection is more difficult than discrete symmetry detection [6] since the condition $f \circ S = f$ must hold for all values of the continuous parameter of $S$. This is corroborated by the increasingly complex methods used to calculate even simple symmetries such as planar rotations [12, 7, 5]. Some methods introduce discretization, where multiple parameter values are chosen and evaluated. LieGAN does this by generating various transformations from the same infinitesimal generator [5]. Introducing discretization increases the complexity of continuous symmetry detection, though any reasonable symmetry detection method must establish whether a continuous symmetry a finite subgroup of a continuous group has been discovered. We believe a vector field approach addresses the issue of discretization. Our vector field-based method reduces the required model complexity of continuous symmetry detection while offering means to detect symmetries beyond affine transformations.

We now provide some background on vector fields and their associated flows (1-parameter transformations). We refer the reader to literature on the subject for additional information [14]. Suppose that $X$ is a smooth[1] (tangent) vector field on $\mathbb{R}^n$:

$$X = \alpha^i \partial_{x^i} := \sum_{i=1}^{n} \alpha^i \partial_{x^i}, \tag{1}$$

where $\alpha^i : \mathbb{R}^n \to \mathbb{R}$ for $i \in [1, n]$, and where $\{x^i\}_{i=1}^n$ are coordinates on $\mathbb{R}^n$. $X$ assigns a tangent vector at each point and can also be viewed as a function on the set of smooth functions which map $\mathbb{R}^n$ to $\mathbb{R}$. E.g. if $f : \mathbb{R}^n \to \mathbb{R}$ is smooth,

$$X(f) = \sum_{i=1}^{n} \alpha^i \frac{\partial f}{\partial x^i}. \tag{2}$$

---

[1]That is, a vector field with $\mathcal{C}^\infty$ coefficient functions.

For example, for $n = 2$, if $f(x, y) = xy$ and $X = y\partial_x$, then $X(f) = y^2$. $X$ is also a *derivation* on the set of smooth functions on $\mathbb{R}^n$: that is, for smooth functions $f_1, f_2 : \mathbb{R}^n \to \mathbb{R}$ and $a_1, a_2 \in \mathbb{R}$,

$$X(a_1 f_1 + a_2 f_2) = a_1 X(f_1) + a_2 X(f_2), \qquad X(f_1 f_2) = X(f_1) f_2 + f_1 X(f_2). \tag{3}$$

These properties are satisfied by derivatives. A flow on $\mathbb{R}^n$ is a smooth function $\Psi : \mathbb{R} \times \mathbb{R}^n \to \mathbb{R}^n$ which satisfies

$$\Psi(0, p) = p, \qquad \Psi(s, \Psi(t, p)) = \Psi(s + t, p) \tag{4}$$

for all $s, t \in \mathbb{R}$ and for all $p \in \mathbb{R}^n$. A flow is a 1-parameter group of transformations. An example of a flow $\Psi : \mathbb{R} \times \mathbb{R}^2 \to \mathbb{R}^2$ is

$$\Psi(t, (x, y)) = (x \cos(t) - y \sin(t), x \sin(t) + y \cos(t)), \tag{5}$$

with $t$ being the continuous parameter known as the flow parameter. This flow rotates a point $(x, y)$ about the origin by $t$ radians.

For a given flow $\Psi$, one may define a (unique) vector field $X$ as given in Equation 2, where each function $\alpha^i$ is defined as

$$\alpha^i = \left( \frac{\partial \Psi}{\partial t} \right) \bigg|_{t=0}. \tag{6}$$

Such a vector field is called the infinitesimal generator of the flow $\Psi$. For example, the infinitesimal generator of the flow given in Equation 5 is $-y\partial_x + x\partial_y$.

Conversely, given a vector field $X$ as in Equation 2, one may define a corresponding flow as follows. Consider the following system of differential equations:

$$\frac{dx^i}{dt} = \alpha^i, \qquad x^i(0) = x_0^i. \tag{7}$$

Suppose that a solution $\mathbf{x}(t)$ to Equation 7 exists for all $t \in \mathbb{R}$ and for all initial conditions $\mathbf{x}_0 \in \mathbb{R}^n$. Then the function $\Psi : \mathbb{R} \times \mathbb{R}^n \to \mathbb{R}^n$ given by

$$\Psi(t, \mathbf{x}_0) = \mathbf{x}(t) \tag{8}$$

is a flow. The infinitesimal generator corresponding to $\Psi$ is $X$. For example, to calculate the flow of $-y\partial_x + x\partial_y$, we solve

$$\dot{x} = -y, \quad \dot{y} = x, \qquad x(0) = x_0, \quad y(0) = y_0 \tag{9}$$

and obtain the flow $\Psi(t, (x_0, y_0))$ defined by Equation 5. It is generally easier to obtain the infinitesimal generator of a flow than to obtain the flow of an infinitesimal generator.

Certain vector fields are particularly significant, having flows which correspond to distance-preserving transformations known as isometries. Such vector fields are known as Killing vectors. See Appendix A and D.4 for details.

We can now connect vector fields and flows with symmetry. A smooth function $f : \mathbb{R}^n \to \mathbb{R}$ is said to be $X$-invariant if $X(f) = 0$ identically for a smooth vector field $X$. The function $f$ is $\Psi$-invariant if, for all $t \in \mathbb{R}$, $f = f(\Psi(t, \cdot))$ for a flow $\Psi$. If $X$ is the infinitesimal generator of $\Psi$, $f$ is $\Psi$-invariant if and only if $f$ is $X$-invariant. If the function $f$ is a machine learning function for a given data set, our strategy is to identify vector fields $X_i$ for which $f$ is invariant. Each vector field $X_i$ for which $f$ is invariant is associated, explicitly or implicitly, with a flow $\Psi_i$, each of which is a 1-parameter subgroup, the collection of which generate the symmetry group. By identifying continuous symmetries by means of $X(f) = 0$, continuous symmetry detection is made to be of similar complexity to that of discrete symmetry detection, since no continuous parameters are present.

The price of this indirect characterization of continuous symmetry is often the loss of the explicit identification of the symmetry group, leaving one with only the infinitesimal generators and the symmetry group merely implied. However, infinitesimal generators can be used to construct an invariant machine learning model as follows. Suppose that $f : \mathbb{R}^n \to \mathbb{R}$ is a (smooth) function of interest for a given data set, and suppose that $X$ is a (smooth) vector field for which $f$ is $X$-invariant. With $X$ fixed, we can consider whether there exist additional functions $\{h^i\}_{i=1}^m$ that are functionally independent of each other (meaning one function cannot be written as a function of the others), each of which are $X$-invariant. If such functions can be found, we can define a new feature space using the functions $\{h^i\}$. That is, we can transform each data point in $\mathcal{D}$ to

the point $(h^1(x_i), h^2(x_i), \ldots, h^m(x_i))$ for each point $x_i$ expressed in the original coordinates $\{x^i\}$. By definition, the transformed data is invariant with respect to the symmetry group, whether that symmetry group is given explicitly or not.

The flow parameter can often be obtained explicitly as well, written as a function of the original coordinates. It is found by solving $X(\theta) = 1$. Where the flow parameters of vector fields can be estimated, our methodology thus inspires symmetry-based feature engineering, with the new features consisting of flow parameters and invariant features.

For example, consider a data set consisting of points $\mathcal{D} = \{x_i\}_{i=1}^r$ in $\mathbb{R}^3$, where the features of each data point are given in Cartesian coordinates $(x, y, z)$. Now suppose that each data point $x_i$ satisfies the equation $f = 0$, where $f = x^2 + y^2 - z$. The function $f$ can be considered to be a machine learning function for this data set, since each point in the data set lies on the level set $f = 0$. We note that $X = -y\partial_x + x\partial_y$ is a vector field for which $f$ is $X$-invariant. With $h^1 = x^2 + y^2$ and $h^2 = z$, the functions $h^1$ an $h^2$ are $X$-invariant. We then construct the data set $\tilde{\mathcal{D}} = \{(h^1(x_i), h^2(x_i)\}_{i=1}^r$. Any machine learning task performed in this feature space will be, by definition, invariant to the flow of $X$, which flow defines the symmetry group of the original data set $\mathcal{D}$.

By estimating vector fields which satisfy $X(f) = 0$, we find seemingly unwanted freedom. This is due to the fact that, assuming $X(f) = 0$, $hX(f) = 0$ for a nonzero function $h$: the question, then, is which of the two symmetries–the flow of $X$ or the flow of $hX$–is "correct." By definition, both are symmetries, so that the function $f$ is symmetric with respect to an apparently large class of symmetries. However, a function is $X$-invariant if and only if it is also invariant with respect to $hX$, so that a set of $X$-invariant functions is also a set of invariant functions of $hX$. Thus, where a vector field $X$ can be used to generate $X$-invariant functions, the result is a feature space which is also invariant with respect to $hX$. This point is further expounded upon in Appendix E.

# 3 Methods

Our method assumes that a given dataset consists of points residing on a differentiable manifold $M$, and that a machine learning function $f$ is a smooth function on $M$. We also assume that the symmetry group we seek is a Lie group generated by 1-parameter transformations known as flows, the infinitesimal generators of which are vector fields which reside in the tangent bundle of $M$.

There are three main stages that define our methodology. The first stage, described in Section 3.1, is the estimation of machine learning functions specific to a dataset. The second and third steps are described in Section 3.2. In the second step, vector fields which annihilate the machine learning functions are estimated. Finally, the vector fields are used to define a feature space which is invariant with respect to the implied flows of the vector fields as described in Section 2. A discussion of known limitations of our methods is included in Appendix C.

## 3.1 Estimating Machine Learning Functions

Probability density functions appear in both supervised and unsupervised problems. The output of a logistic regression model or, similarly, a neural network model that uses the softmax function, can be interpreted as estimating the probability that a given point belongs to a particular class. In fact, the probabalistic outputs of many other machine learning models can be analyzed for symmetry, provided that the probability function is differentiable. Invariant features of the symmetries of these functions define level sets upon which the probability is constant. If the flow of an infinitesimal generator can be given, a point in the dataset can be transformed to another point–not generally in the given dataset–and the model should predict the same probability for the transformed point as it would for the original point. Applications include supervised feature engineering and model validation, where predictions on generated points can be used to assess a model's ability to generalize.

Particularly in the unsupervised case, density estimation can be challenging, complicating methods that require their estimation. We propose to address this potential shortcoming of density-based symmetry detection by a method we call *level set estimation*. Level set estimation seeks to learn a function $F : \mathbb{R}^n \to \mathbb{R}^k$ $(k < n)$ such that $F(x_i) = \mathbf{0}$ for each $x_i \in \mathcal{D}$. Any embedded submanifold can be, at least locally, expressed as a level set of a smooth function [14], and so level set estimation operates on the assumption that the data points belonging to a given dataset lie on an embedded

submanifold. If such a function can be estimated for a given data set, one can estimate symmetries of the component functions $f_i$ of $F$.

Level set estimation can be done as follows, although we note that further development of level set estimation may lead to more effective and/or efficient implementations. First, we construct a function $f$ as a linear combination of pre-determined features, as with a generalized additive model or polynomial regression, with model coefficients $\{a_i\}_{i=1}^m$. An $m$-component column vector $w$ is constructed using the coefficients $a_i$. Then, for each point in the dataset, we set the target to 0, so that a function $f$ is approximated by estimating

$$Bw = \mathcal{O}, \tag{10}$$

where $B$ is the feature matrix consisting of $m$ columns and $N$ rows, where $N$ is the number of points in the dataset, and where $\mathcal{O}$ is the zero matrix. OLS regression motivates the way in which we solve Equation (10), though we note that a solution $w_0$ is equivalent to a scaling $cw_0$ for some constant $c$. We therefore turn to constrained regression, choosing, by default, the constraint $\sum_{i=1}^m a_i^2 = 1$. Additionally, we note that any loss function can be used, with predictions and targets determined by the left- and right-hand sides of Equation (10), respectively.

We can estimate more than a single component function $f$ by appending additional columns to $w$, in which case we constrain the columns of $w$ to be orthonormal. As appending columns to $w$ is likely to increase any cost function used to estimate $w$, we use an "elbow curve" to determine when the number of columns of $w$ is sufficiently high. That is, in the presence of a sudden comparatively large increase in the cost function as the number of columns of $w$ is increased, the number of columns of $w$ is chosen as the value immediately before the large increase is encountered. We handle the constrained optimization problem of (10) using existing constrained optimization algorithms [15, 16].

The freedom of expression in the components of the level sets is, unfortunately, not fully contained by constrained optimization. When certain functions are chosen as part of the level set estimation, such as polynomials, this scaling issue can extend to multiplication by functions. For example, consider a data set in $\mathbb{R}^3$ known a priori to be approximately described by the function $F = \mathbf{0}$, where the components of $f_1$ and $f_2$ of $F$ are given as

$$f_1 = x^2 + y^2 - 1, \qquad f_2 = z - 1. \tag{11}$$

Such a level set could also, at first glance, be described by the following equations:

$$x^2 + y^2 - 1 = 0, \qquad z - 1 = 0, \qquad x(z-1) = 0. \tag{12}$$

From a mathematical point of view, the third equation is degenerate and does not change the intrinsic dimension of the manifold characterized by this level set, owing to the fact that the rank of this function when $x = 0$ is 2. However, within the context of learning level sets from data, the extrinsic dimension may be large, and it may not be practical to examine the components of the learned level set. Furthermore, key points for determining the rank of the corresponding function may not be present in the data.

A potential workaround for this problem is to incrementally increase the allowed degree of the learned polynomial functions in Equation 12. For the example considered in Equation 11, we can first optimize over the space of degree 1 polynomials, ideally discovering a scaling of the function $f_2$. Next, when searching for quadratic components, we can extend the columns of $w$ which correspond to any allowed terms of the form $h(x, y, z)f_2$.[2] With constrained optimization, the column space of $w$ can then be expressed as an orthonormal basis. The optimization routine can then continue, with the final level set model for $w$ not including the artificially-created columns. Alternatively, if one is interested primarily in at most quadratic component functions, one can first optimize using affine functions, then project the data onto the lower-dimensional hyperplane defined by the affine components, subsequently searching for quadratic terms on the reduced space where no additional affine components should be present. We provide experiments which apply these concepts in practice.

One may well ask whether level set estimation can be accomplished using a neural network rather than constrained polynomial regression. Besides the problem of uniqueness just spoken of, a neural network approach would likely require plentiful off-manifold examples so as not to learn a function which is identically zero. A neural network approach to level set estimation is not explored in this paper, though may be explored in the future.

---

[2]At degree 2, $h$ could be, for example, $x$, but not $x^2$, since $x^2z$ is not a degree 2 polynomial.

The last type of machine learning function we consider is a metric tensor. Given a (pseudo-) Riemannian manifold with a metric tensor, we can calculate infinitesimal isometries of the associated (pseudo-) Riemannian manifold. We provide a means for approximating the metric tensor in certain cases in Appendix A.

## 3.2 Estimating the Infinitesimal Generators and the associated Invariant Feature Space

With scalar-valued machine learning functions $f_i$, we construct a vector-valued function $F$ whose components are defined by the functions $f_i$. We can then obtain vector fields which annihilate the components of $F$ by calculating nullspace vectors of $J$, the Jacobian matrix of $F$.[3] More vector fields may annihilate the functions than can be identified by the nullspace of $J$, though our method is inspired by this idea. First, we construct a vector field whose components are an arbitrary linear combination of pre-determined features, as in a general additive model, though we primarily use polynomial features herein. The coefficients of this linear combination reside in a matrix $W$, with the columns of $W$ corresponding to the coefficients for a single vector field. We then estimate $W$ in the following equation:

$$MW = \mathcal{O}, \tag{13}$$

where $M = M(\mathcal{J}, B)$ is the *Extended Feature Matrix* computed using the array $\mathcal{J}$ of Jacobian matrices at each point, $B$ is the feature matrix, and $\mathcal{O}$ is the zero matrix. The matrix $M$ is computed via Algorithm 1.

---

**Algorithm 1** Constructing the Extended Feature Matrix $M$

---

$m \leftarrow$ number of features
$n \leftarrow$ dimension of space
$N \leftarrow$ number of points in the dataset
$\mathcal{J} \leftarrow Jacobian(F)(x_i)$           ▷ 3-d array of Jacobian matrices at each point
**for** i in range($N$) **do**
     $row_0 \leftarrow padded(B_i)$           ▷ m(n-1) 0's are appended to the $i^{th}$ row of $B$
     **for** j in range(1,m-1) **do**
         $row_j \leftarrow roll(row_0, j \cdot m)$     ▷ Each subsequent row is displaced $m$ entries to the right.
     **end for**
     $b_i \leftarrow Matrix([row_0, \dots row_{m-1}])$
     $mat_i \leftarrow \mathcal{J}_i b_i$           ▷ Multiply the Jacobian of $F$ at $x_i$ by the matrix $b_i$
**end for**
$M \leftarrow StackVertical([mat_0, \dots mat_{N-1}])$         ▷ This matrix has size $nN \times nm$.

---

As with Equation (10), a solution to Equation (13) is estimated using a constrained optimization of a selected loss function, for which we turn to *McTorch* [16]. Using this pytorch-compatible library, one can choose from a variety of supported loss functions and (manifold) optimization algorithms. As with level set estimation, we sequentially increase the number of columns of $W$ until an "elbow" in cost function values is reached.

However, we find the same unwanted freedom as with level set estimation, since $X(f) = 0$ implies that for a smooth function $h$, $hX(f) = 0$. Since we are assuming that the components of $X$ are polynomial functions, it is possible that a such a pair $X$ and $hX$ may exist in the search space. The means of dealing with this unwanted freedom can depend on the particular experiment: if there are no affine symmetries, for example, vector fields with components of degree 2 can be approximated without regard to this issue.

To define a model which is invariant with respect to the flows of pre-computed vector fields $X_i$, we seek features $h^j$ (separate of the functions $f_i$) such that $X_i(h^j) = 0$. This we do by estimating $v$ in the following equation:

$$M_2 v = \mathcal{O}, \tag{14}$$

where $M_2 = M_2(\mathcal{J}, B, \mathcal{B})$ is the *Invariant Function Extended Feature Matrix* computed via Algorithm 2, $\mathcal{B}$ is the feature matrix of $W$ (already estimated), and $B$ is the feature matrix of the unknown function.

---

[3]In fact, if $F$ is given in closed form, this can be done using symbolic software and without referencing the original data.

---

**Algorithm 2** Constructing the Invariant Function Extended Feature Matrix $M_2$

---

$m \leftarrow$ number of features
$n \leftarrow$ dimension of space
$N \leftarrow$ number of points in the dataset
$\bar{J}(B)(x_i) \leftarrow$ Jacobian matrix of features of $B$ at $x_i$      ▷ size $m \times n$: derivatives of raw features.
$\mathcal{J} \leftarrow [\bar{J}(B)(x_i)]$      ▷ Array of Feature Jacobian matrices at each point.
$M = M(\mathcal{J}, \mathcal{B})$      ▷ Extended Feature matrix computed via Algorithm 1
$M_2 = MW$      ▷ This is not $\mathcal{O}$, since the arguments of $M$ are different than in 13

---

This method assumes that the invariant functions can be expressed in terms of prescribed features, which, in our case, are typically polynomial functions. The quantity $v$ can be estimated from Equation 14 in much the same way that we estimate other quantities, in that we constrain $v$ to be orthogonal, then optimize a selected loss function using *McTorch* [16]. We should note that while this method may produce invariant features, they are not guaranteed to be functionally independent. For example, in our polynomial feature space, the vectors corresponding to $x$ and $x^2$ are orthogonal, though the functions themselves are not functionally independent.

### 3.2.1 Comparison of Discovered Symmetries to Ground-Truth Symmetries

To evaluate symmetry discovery methods in general, it is important to quantify the ability of a given method to recover ground truth symmetries in controlled experiments. For a ground truth vector field $X = \sum_{i=1}^{N} f_i \partial_{x^i}$ and an estimated vector field $\hat{X} = \sum_{i=1}^{N} \hat{f}_i \partial_{x^i}$, our similarity score between them is defined to be

$$\text{sim}(X, \hat{X}) = \frac{1}{N} \sum_{i=1}^{N} \frac{|\langle f_i, \hat{f}_i \rangle|}{||f_i|| \cdot ||\hat{f}_i||}, \tag{15}$$

where

$$\langle f_i, \hat{f}_i \rangle = \int_{\Omega} f_i \hat{f}_i d\mathcal{M}, \qquad ||f_i|| = \sqrt{\langle f_i, f_i \rangle},$$

and with $\Omega$ being defined by the range of a given dataset–herein, we take the full range of the dataset. We note that under a change of coordinates induced by a diffeomorphism, the similarity is invariant, owing to the Jacobian which would be present in the transformed definite integrals. Further refinement of this similarity score is left as future work.

## 4 Experiments

Our first experiment will compare our method to LieGAN in an effort to discover an affine symmetry. Our next experiment will demonstrate our method's ability to identify symmetries which are not affine. The third experiment demonstrates that symmetry can occur in real data, and that our method can detect approximate symmetry in real data. Our last main experiment deals with pre-determined models which are not polynomials–in particular, when the ground truth symmetry is not expressible in terms of polynomial coefficient functions. Additional experiments appear in Appendix D: some highlights are symmetry of a model-induced probability distribution, discrete symmetry estimation, and estimation of infinitesimal isometries.

### 4.1 Affine Symmetry Detection Comparison

Our first experiment compares the performance of our method to LieGAN, which is the current state of the art in symmetry detection. We use simulated data where the "true" symmetries are known. Our three datasets for this experiment consist of $N$ random samples generated from a 2-dimensional normal distribution with mean $(1, 1)$ and covariance matrix $\begin{bmatrix} 4 & 0 \\ 0 & 1 \end{bmatrix}$, where $N \in \{200, 2000, 20,000\}$. In each case, target function values are generated as $f = (x - 1)^2 + 4(y - 1)^2$ for each point $(x, y)$ in the dataset. The function exhibits symmetry: $X(f) = 0$, with $X = -(4y - 4)\partial_x + (x - 1)\partial_y$ (or a scaling thereof). In the methodology employed by LieGAN, the ground truth infinitesimal generator is expressible in terms of a $3 \times 3$ matrix, with generated transformations acting on the $z = 1$ plane in three dimensions. Thus, in the case of LieGAN, the data is generated with mean $(1, 1, 1)$ and covariance given with the upper $2 \times 2$ matrix as before, with all other values besides the bottom right

Table 1: Affine Symmetry Detection Comparison on a Gaussian Distribution

| $N$ | LieGAN similarity | Our similarity | LieGAN time (s) | Our time (s) |
|---|---|---|---|---|
| 200 | $0.7636 \pm 0.2862$ | $1.0000 \pm 5.6 \cdot 10^{-7}$ | $2.1717 \pm 0.0481$ | $2.1778 \pm 0.0125$ |
| 2000 | $0.9990 \pm 0.0014$ | $1.0000 \pm 1.7 \cdot 10^{-7}$ | $18.1289 \pm 0.2528$ | $2.6412 \pm 0.0208$ |
| 20000 | $1.0000 \pm 1.5 \cdot 10^{-5}$ | $1.0000 \pm 3.2 \cdot 10^{-8}$ | $195.5748 \pm 2.6080$ | $5.6177 \pm 0.0585$ |

value being 0, which bottom right value is $0.0001$ so as to approximately lie on the $z = 1$ plane. Our own method uses the 2-dimensional data.

Our method first employs polynomial regression to approximate the function $f$ from the data. Then, with a model using affine coefficients for the estimated vector field, we estimate $W$ in Equation (13) using the $L_1$ loss function and the Riemannian Adagrad optimization algorithm [16] with learning rate $0.01$ for 5000 epochs.

We compare the results of our method to the results of LieGAN using two metrics, namely similarity to the ground truth defined by Equation (15) and computation time. The results are shown in Table 1 and use 10 trials for each $N$. These results suggest that if a sufficient amount of data is given, LieGAN is able to correctly identify the ground truth symmetry. However, our method exhibits a clear advantage for $N = 200$ while offering comparable similarity scores to LieGAN for larger sample sizes. Perhaps the most striking feature of Table 1, however, is the dramatic difference in time needed to correctly identify the symmetry. Our method offers a clear advantage in terms of computation time as the number of samples increases.

## 4.2 Non-Affine Symmetry Detection Comparison

We now demonstrate the ability of our method to discover symmetries which are not affine. We generate 2000 data points from a normal distribution with mean $(0, 0)$ and covariance matrix $\left[\begin{smallmatrix} 4 & 0 \\ 0 & 4 \end{smallmatrix}\right]$. For each data point, we generate target values according to $f = x^3 - y^2$. A ground truth vector field which annihilates this function is $2y\partial_x + 3x^2\partial_y$.

We first use polynomial regression to approximate $f = x^3 - y^2$. Then, we estimate $W$ in Equation (13) using the $L_1$ loss function, selecting the Riemannian Adagrad optimization algorithm [16] with a learning rate of $0.1$, training for 5000 epochs. Our method then obtains a similarity score of $0.9983 \pm 0.0006$, having aggregated across 10 independent trials. Meanwhile, LieGAN obtains a similarity score of $0.0340 \pm 0.0142$ on 10 independent trials, being unable to discover symmetries of this complexity.

## 4.3 Symmetry Detection using Realistic Data

Our next experiment uses real data that is publicly available, which dataset we refer to as the "Bear Lake Weather" dataset [17]. The dataset gives daily weather attributes. It contains $14,610$ entries with 81 numeric attributes including the daily elevation level of the lake. The dataset contains precisely 40 years' worth of data from October of 1981 through September of 2021.

We believe an understanding of the behavior of the weather in the past is relevant to this problem. Therefore, we first construct time series of length in dimensions by means of a sliding window of length days: the first time series is constructed using the first 1461 days (the number of days in four years). The next time series is constructed using the second day through day 1462, and so forth.

After converting the raw data to time series data, we apply a transformation on the data meant to extract time-relevant features of the data known as the Multirocket transform [18]. We select 168 kernels in our transform. The Multirocket transform transforms the data from $13,149$ time series of length 1461 in 81 variables to tabular data: $13,149$ entries in 1344 variables.

For such high-dimensional data, we turn to PHATE, a state of the art data visualization method [19]. Using PHATE, we reduce the dimension to 2, so that our new dataset has $13,149$ entries in 2 variables. The resulting data appears to approximately lie on a circular shape and is shown in Figure 1.

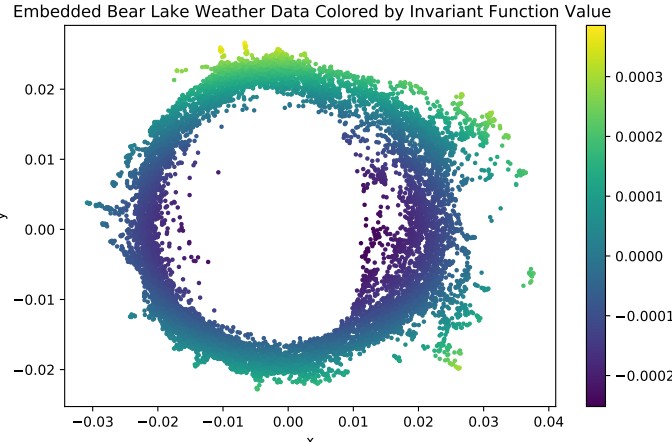

Figure 1: The ROCKET-PHATE embedded Bear Lake Weather dataset, colored by invariant function value.

Figure 1 suggests that the data is largely periodic. In fact, further experimentation reveals that the points for a given calendar year correspond to a traversal around the circular-like shape. Thus, for the analysis of non-seasonal weather patterns, it may be of use to examine features of this embedded dataset which are invariant under the periodic transformation, the approximate symmetry.

Indeed, our method reveals an approximate invariant function given by

$$0.33592x^2 + 0.94189y^2 - 2.9743 \cdot 10^{-4} = 0.$$

This result was obtained using level set estimation, assuming a function expressible as $ax^2 + by^2 + c$, using the Mean Square Error loss function with Equation (10), optimized using Riemannian stochastic gradient descent with a learning rate of $0.001$ and trained for $5000$ epochs. This experiment shows that symmetry can occur in real data, and that our method can detect symmetry and estimate invariant functions for real data.

### 4.4 Symmetry Detection with Non-Polynomial Ground Truth

This experiment deals with a case in which the ground truths are not strictly polynomial functions. We generate 2048 numbers $x_i$ and 2048 numbers $y_j$ each from $U(0, 2\pi)$. Next, for each pair $(x_i, y_i)$, we obtain $z_i$ by means of $z_i = \sin(x_i) - \cos(y_i)$, so that a ground-truth level set description of the data is given as $z - \sin(x) + \cos(y) = 0$.

We first apply our level set estimation method to estimate this level set. We optimize the coefficients of the model

$$a_0 + a_1 x + a_2 y + a_3 z + a_4 \cos(x) + a_5 \cos(y) + a_6 \cos(z) + a_7 \sin(x) + a_8 \sin(y) + a_9 \sin(z) = 0$$

subject to $\sum_{i=0}^{9} a_i^2 = 1$. In light of Equation 10, the matrix $B$ has a row for each of the 2048 tuples $(x_i, y_i, z_i)$, and 10 columns, which columns correspond to the 10 different feature functions in our pre-determined model. The vector $w$ contains all 10 parameters $\{a_i\}_{i=0}^9$.

Using the $L_1$ loss function and the Riemannian Adagrad optimization algorithm [16] with learning rate $0.01$, our estimated level set description is

$$-0.57737z - 0.57713 \cos(y) + 0.57756 \sin(x) = 0,$$

which is approximately equivalent to the ground truth answer up to a scaling factor.

Having demonstrated that our method easily extends to non-polynomial pre-determined features, we now wish to examine a case in which a polynomial pre-determined model of symmetry does not contain the ground truth vector field. For this, we use the same dataset, albeit we define $f_i = z_i$, so

that we are approximating symmetries of $f(x, y) = \sin(x) - \cos(y)$. A ground truth vector field which characterizes the symmetry of $f$ is given as

$$X = \sin(y)\partial_x - \cos(x)\partial_y.$$

Applying our method with a pre-determined model of degree 2 polynomial coefficients gives an estimated (using Equation 13) vector field $\hat{X}$ of

$$\hat{X} = \left(0.7024 - 0.1874x - 0.2203y + 0.0121x^2 + 0.0242xy + 0.0133y^2\right)\partial_x$$

$$+ \left(-0.5783 + 0.2665x + 0.1236y - 0.0311x^2 - 0.0150xy - 0.0097y^2\right)\partial_y.$$

This result was obtained using the $L_1$ loss function, the Riemannian Adagrad optimizer [16] with learning rate 0.1, training for 5000 epochs. We also note that $\text{sim}(X, \hat{X}) = 0.62$, using the similarity score defined in Equation (15). This demonstrates that our method can be applied even when a ground truth vector field is not covered by the vector field pre-determined model.

## 5   Conclusion

While current state of the art experiments focus, in the context of continuous symmetry detection, on affine symmetries, we have outlined a method to estimate symmetries including and beyond the affine symmetries using vector fields. Our method relies on estimating machine learning functions such as a probability distribution, continuous targets for regression, or functions which characterize an embedded submanifold (level set estimation). Our method also requires, in the case of discrete symmetry detection, that the transformation (sub)-group be parameterized, so that the "best-fit" parameters can be obtained by optimization. In the case of continuous symmetry detection, vector fields are constructed as arbitrary linear combinations of a chosen basis of vector fields, and the parameters that are the coefficients of the linear combination are optimized. In both discrete and continuous symmetry detection, the parameter space is often constrained. Herein, we have chosen to express the coefficients of our vector fields in terms of polynomial functions.

When searching for affine symmetries in low dimensions, our method is appealing when compared with current state of the art methods. This is due primarily to the relative ease of implementation. Additionally, when compared with the state of the art, our method appears to offer a computational advantage while at least competing and often outperforming in terms of accuracy, especially for more complex symmetries than affine ones.

Our infinitesimal generators are a step removed from the symmetry group than what is typical of symmetry detection methods. We have proposed building invariant architectures based on the identification of features which are invariant with respect to the discovered vector fields, and we have demonstrated the estimation of these invariant features. This results in an invariant model, since the information fed to the model is approximately invariant with respect to the symmetry group.

We have also discussed potential novel applications of symmetry discovery, including model validation and symmetry-based feature engineering. Further potential applications are discussed in Appendices A and D.4. Our novel approach for detecting symmetry in data, the computational advantages of our method, and the demonstrated and potential applications of our method make our method of interest to the machine learning community.

Future work includes the following. First, we plan to extend our method to express the vector fields in greater generality beyond polynomial functions. A natural extension of this would be to express the invariant features in greater generality as well. Additionally, as discussed in the appendices, the detection of isometries relies on the ability to compute Killing vectors, for which, in the case of non-Euclidean geometries, no machine-learning compatible methods have been apparently implemented. Also discussed in the appendices is the notion of metric tensor estimation, which we leave, in large part, as work for the future.

## Acknowledgments

This research was supported in part by the NSF under Grants 2212325 [K.M.], CIF-2212327 [A.M.], and CIF-2338855 [A.M.].

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

# A Killing Vectors, Isometries, and Metric Tensor Estimation

Certain vector fields have flows corresponding to isometric transformations, i.e. a transformation which preserves manifold distances. Such infinitesimal isometries are called Killing vectors, named after Wilhelm Killing. The search for Killing vectors can be done in much the same way as with ordinary vector fields, although instead of searching the parameter space defined by arbitrary polynomial coefficients, one constructs an arbitrary linear combination of the Killing vectors of the space of interest, optimizing the coefficients.

However, this method requires knowing the Killing vectors for the space of interest. In general, Killing vectors can be calculated if a Riemannian (or pseudo-Riemannian) metric is given [20]. In machine learning, data with continuous-valued features are usually assumed not only to lie in $\mathbb{R}^n$, but on the Riemannian manifold $\mathbb{R}^n$ equipped with the Euclidean metric. Within this context, the Killing vectors form a proper subspace of the generators of the affine transformations, and are thus obtainable by previous methods. Nevertheless, our methods have the following advantages over existing ones. First, our methods can detect the infinitesimal generators of affine transformations and, as our experiments suggest, may offer a large computational advantage in certain situations to existing methods. Second, the assumption that a suitable Riemannian metric on the data space is the Euclidean metric appears to be an assumption only. Generally, inner products of (tangent) vectors are computed using the dot product without consideration of structure which may naturally exist in the data. In fact, one discovered symmetry of LieGAN is an infinitesimal generator corresponding to a subgroup of the Lorentz group [5], which generator is an isometry of the Lorentz metric and not the Euclidean metric. In that particular context, the Euclidean assumption is replaced with the assumption that the (pseudo-Riemannian) metric is the Lorentz metric, which assumption is appropriate given the relation to physics[4] This point is nontrivial, since the decision of which Riemannian metric to endow a differentiable manifold with impacts not only which vector fields generate the isometry group, but also suggests the way in which (tangent) vectors should be "dotted."

We are then led to the question of what assumptions can be made about the form of the Riemannian metric for a data space modeled as lying on a Riemannian manifold. This question is not fully addressed herein, though we will describe a practical scenario in which the Riemannian metric can be assumed to be non-Euclidean.

Consider a data set $\mathcal{D}$ whose data points lie in $\mathbb{R}^n$. There are several dimensionality-reduction techniques that may seek to map points in $\mathcal{D}$ to form a data set $\tilde{\mathcal{D}}$ whose points lie in $\mathbb{R}^m$, where $m < n$. These techniques are thought to provide a mapping $\Phi : \mathbb{R}^n \to \mathbb{R}^m$. However, such an algorithm provides a bijective mapping from the finite set $\{x_i\}$ ($x_i \in \mathcal{D}$) to the finite set $\{\tilde{x}_i\}$ ($\tilde{x}_i \in \tilde{\mathcal{D}}$). Therefore, we can seek to learn functions $x_i(\tilde{x})$ via regression, then use the learned functions as the components of a function $\Theta : \mathbb{R}^m \to \mathbb{R}^n$. Insofar as the Euclidean metric assumption is valid on $\mathbb{R}^n$, we can obtain a metric $g$ on $\mathbb{R}^m$ by means of the so-called "pullback" of $\Theta$, then calculate the Killing vectors of $g$: this allows one to calculate the infinitesimal isometries of a distribution or level set on $\mathbb{R}^m$, the components of which will not generally be affine functions.

Obtaining a metric tensor by means of $\Theta$ operates under the assumption that the data points $x_i$ lie on a manifold of dimension $m$ embedded in $\mathbb{R}^n$, and that the coordinates of the data points $\tilde{x}_i$ are suitable manifold coordinates, at least locally. The strength of the model metric tensor thus depends on the extent to which these assumptions are valid. An example of a relatively strong model would be to begin with a data set in $\mathbb{R}^n$, construct a new data set $\mathcal{D}$ consisting of points in $\mathbb{R}^n$ which points are projected onto the first $m$ principal components, and then let $\tilde{\mathcal{D}}$ be the data set constructed by writing each point in $\mathcal{D}$ in the basis defined by the $m$ principal components. An example of a relatively weak model would be to use the coordinates $(x, y)$ to describe points in $\tilde{\mathcal{D}}$, where the points in $\mathcal{D}$ lie on the surface of a sphere in $\mathbb{R}^3$.

We should make a couple of points clear at this point. First, if one wishes, the Killing vectors (any smooth vector field, for that matter) of the metric tensor $g$ can be expressed as vector fields on $\mathbb{R}^n$: that is, in terms of the original coordinates. In the original coordinates, the flow of any Killing vector will represent an affine transformation, since the metric on $\mathbb{R}^n$ is the Euclidean metric, begging the question of whether the symmetry should have been estimated in the original coordinates where only a search among affine symmetries is needed. One potential reason for this is due to the curse

---

[4]For full details, we refer the reader to [5].

of dimensionality: as our estimation of infinitesimal generators uses (constrained) regression, the performance of our methods may suffer in higher dimensions, suggesting that, in certain cases, it may be more practical to estimate more complicated symmetries in lower dimensions than simpler symmetries in higher dimensions. However, this point requires experimentation in addition to the experiments conducted herein.

The second point we wish to make clear is that the procurement of Killing vectors for a given metric is no small task. In fact, generic Riemannian metrics admit few, if any, Killing vectors: the defining equations for Killing vectors form an overdetermined system of differential equations[5]. On that point, while the python library *geomstats* [21] offers the ability to estimate the metric tensor using a function such as $\Theta$, it appears that the estimation of Killing vectors is not yet supported. Thus, our experiment which seeks to identify isometries as described in this section uses symbolic software to compute the Killing vectors [22].

Metric tensor estimation has the potential to make a large impact on the way in which certain aspects of machine learning are approached. Apart from adjusting the metric tensor according to transformations as discussed above, metric tensor and Killing vector estimation could impact manifold alignment. In general, manifolds whose points are put into bijective correspondence (in a smooth manner) should admit the same number of Killing vectors. Estimating the infinitesimal isometries may thus help to bring added clarity to manifold alignment. And, of course, a choice of a non-Euclidean metric tensor will have a large impact on the way inner products and manifold distances are computed.

## B  Similar Methodology for Discrete Symmetries

While much of this paper focuses on the identification of continuous symmetry, we also present an analogous method by which discrete quantity-preserving transformations can be identified. Given a quantity of interest denoted by the function $f$, a transformation $S$ is said to define a symmetry of the data if $f(x_i) = f(S(x_i))$ for each point $x_i$ in the data set. For example, if $f = y - x^2$ and the equation $f = 0$ is satisfied for each point in the data set, the data set is symmetric under a reflection about the line $x = 0$, which is a discrete transformation.

To estimate discrete symmetries, we assume that a symmetry belongs to a parametric family of transformations, which parameters we denote by $\mathbf{w}$. We define, for each $x_i$ in a given dataset,

$$\hat{y}_i = S(f(x_i); \mathbf{w}) - f(x_i), \qquad y_i = 0, \tag{16}$$

and we seek to minimize $L(\hat{y}_i, y_i)$ for a smooth loss function $L$, typically by means of constrained optimization. For example, we consider the 2-dimensional task of determining a line of reflection about which a function $f(x, y)$ is symmetric. The reflection of a point $(x, y)$ about a line in the form of $ax + by = 0$ can be written as

$$S(x, y; a, b) = \frac{1}{a^2 + b^2} \begin{bmatrix} a^2 - b^2 & -2ab \\ -2ab & b^2 - a^2 \end{bmatrix} \begin{bmatrix} x \\ y \end{bmatrix}. \tag{17}$$

Minimizing $L(\hat{y}_i, y_i)$ for a smooth loss function then estimates the parameters $(a, b)$ which characterize the line for which $f(x, y)$ is approximately reflectively symmetric. Since the line $ax + by = 0$ is equivalent to $cax + cby = 0$ for some constant $c$, our method is best accomplished by means of constrained optimization, which, in this example, is easily accomplished by means of $a^2 + b^2 = 1$.

For continuous symmetries, invariant quantities beyond those found to be of significance to the data set itself–the functions $h^i$–were used to define an "invariant feature space." That is, a set of features were given such that when the data points were expressed in terms of these features, any machine learning task performed on this new feature space would necessarily be invariant with respect to the given symmetry group. For discrete symmetry detection, the transformations can be given explicitly, so that existing methods for building invariance into a model that require the symmetry group to be given beforehand can be employed. However, it may also be possible–perhaps depending on the discovered symmetries–to calculate features $h^i$ that are invariant with respect to the discovered discrete symmetries. For example, the function $h^1 = y$ is invariant to reflection about the $x = 0$ line.

---

[5]We should also note that efforts to compute Killing vectors without solving the Killing equations directly have been successful[20]

The identification of features which are invariant with respect to a fixed collection of symmetries would allow one to transform points from the original feature space to the invariant feature space, so that any machine learning task would be invariant with respect to the given symmetry group.

We also note that our assumption that the discrete symmetries sought originate in parametric families of transformations is limiting. In particular, despite previous work conducted with permutation groups [6], it does not appear that our method can be applied to permutation groups.

## C   Limitations

Our method is an improvement to state of the art methods for continuous symmetry detection, both for computational reasons and due to the fact that our method can detect continuous symmetries which are not affine transformations. However, there are limitations to our approach. First, as we have mentioned previously, is the issue which arises when vector fields $X$ and $fX$ are both present in the search space. We have suggested that a symbolic approach may resolve this, though non-symbolic approaches such as ours must grapple with this issue. Our approach of searching for polynomials of lower order first partially resolves this issue. Another promising resolution is to restrict the search space of vector fields to particular types of symmetries, such as isometries, which is experimented with in Appendix D.4.

Another limitation is that our method requires an assumption about the coefficient functions of the vector fields. Herein, we primarily use polynomial functions, and although polynomials are universal approximators [23], the number of polynomial terms for a given coefficient function can increase rapidly, particularly when the number of dimensions is high. We believe this limitation is best resolved, as with the previous limitation, by restricting the search space to particular types of symmetries.

Though our method is comparatively efficient, the reliance on the elbow curve to determine both the number of discovered symmetries and, when level set estimation is used, the number of coefficients in the level set, is a limitation. This limitation is of a more persistent nature than the previous limitations, appearing in analogous form as the dimension hyperparameter in manifold learning methods, or as the number of clusters in $k$-means clustering. It is possible that our method would be improved by another means of selecting this number, however.

We also mention the limitations inherited via the choice of optimization of equations (13) and (14). Cost function selection, optimization algorithm selection, hyperparameter selection, together with the potential for poor parameter initialization all constitute limitations to our method. We believe the optimization task is much more straight-forward than training current state of the art symmetry detection models, however, as other methods require fine-tuning neural networks.

Lastly, our notion of similarity of vector fields, while adequate for our experiments, is limited. Primarily, when a collection of vector fields is estimated and taken to constitute the infinitesimal generators for the symmetry group, it is not clear how to compare the estimated collection of vector fields to a ground truth collection. The similarity score also depends on the range of a given dataset, and the aggregation of component-wise similarities via the arithmetic mean is naïve, leaving room for further refinement. In particular, in the presence of a metric tensor $g$, one may seek to define the similarity score by means of $\mathbb{E}\left[\langle X, \hat{X}\rangle_g\right]$, where the inner product $\langle X, \hat{X}\rangle_g$ is defined by the metric $g$ on the tangent space of the manifold, and where the expected value is computed by means of the underlying distribution of the data. However, exploration of such refinements are left as future work.

## D   Additional experiments

### D.1   Invariant Feature Construction

We have claimed that, given vector fields $X_i$, one can find a certain number of features $h_j(\mathbf{x})$ such that $X_i(h_j) = 0$. For a single vector field, it is generally true that $n - 1$ functionally independent functions can be found, where $n$ is the dimension of the space. However, when more vector fields are considered, the number of functions invariant with respect to all vector fields decreases. Additionally, it is not clear whether, for a given machine learning task, it is better to identify invariant features or features which correspond to the flow parameters of the vector fields: for example, the angle $\theta$

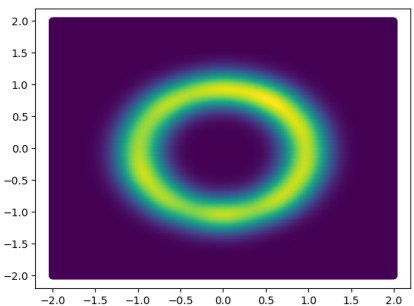 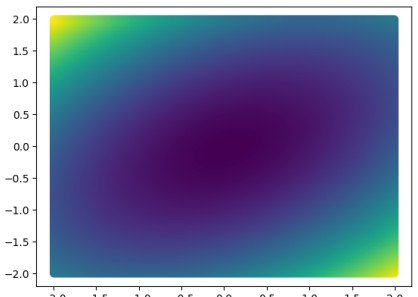

Figure 2: A visual representation of the kernel density estimate of the circular dataset.

Figure 3: A representation of the estimated invariant function of the symmetry of the circular probability distribution.

corresponds to the flow parameter of a rotational transformation, while the radial feature is invariant–a given machine learning task may favor one above the other.

Despite the apparent challenges with both estimation and interpretation, we will estimate an invariant feature in the simplest possible example: we will try to recover the radial feature for a circular dataset. The data is generated by uniformly sampling $\theta$ randomly on $[0, 2\pi)$. Then, we choose $x = \cos(\theta)$ and $y = \sin(\theta)$. The probability distribution is estimated using kernel density estimation, and a single vector field is estimated by applying $X(p) = 0$, where $p$ is the probability distribution, and where the components of $X$ are assumed to be affine. Finally, we optimize $X(h) = 0$, where we assume $h$ takes the form

$$h = a_1 x^2 + a_2 y^2 + a_3 xy + a_4 x + a_5 y + a_6,$$

with $\sum_{i=1}^{6} a_i^2 = 1$. After estimating $h$, we plot a grid of function values of $h$. The result is pictured alongside the visualization of the estimated probability distribution of the data (Figs. 2 and 3).

### D.2 Level Set Estimation

We now seek to find an invariant function of a circular dataset using level set estimation. As alluded to previously, certain datasets may have an underlying distribution which is not symmetric, and yet the data points themselves may lie on an embedded surface which does exhibit symmetry.

We return again to the circle, though we initially embed the circle in three dimensions. We sample 2000 values $t_i$ from a standard normal distribution, then obtain angles $\theta_i$ via $\theta_i = t_i \bmod 2\pi$. We then obtain 3-dimensional data by means of the following:

$$x = \cos(\theta), \qquad y = \sin(\theta), \qquad z = 1. \tag{18}$$

As noted previously, level set estimation with polynomials can lead to degenerate expressions for the level set. Therefore, we initially restrict our search space to affine functions. We optimize according to 10 sequentially, first taking the number of (orthonormal) columns of $W$ to be 1, then 2. The chosen loss function value with the number of columns being 1 is approximately $7 \cdot 10^{-12}$, while the loss function value with two columns is approximately $7 \cdot 10^{-2}$. Therefore, we choose $W$ to have one column, which approximately represents the following function, ignoring the $x$ and $y$ coefficients, which are less that $10^{-5}$:

$$-0.70710 + 0.70710z = 0. \tag{19}$$

We wish to continue looking for suitable functions that, when combined with 19, describe the manifold on which the data lies. Thus, we will expand our search space to quadratic functions. Before doing so, in an attempt to avoid discovering equations degenerate with respect to 19, we will project the 3-dimensional data onto the space defined by our discovered affine functions–in this case, the approximate $(x, y)$ plane (though technically $z = 1$). A plot of the result is obtained in 4. We then apply 10 on the reduced space, now with quadratic features. The discovered component is given as

$$0.5779y^2 - 0.0044xy + 0.5771x^2 + 0.0007y - 0.0018x - 0.5770 = 0, \tag{20}$$

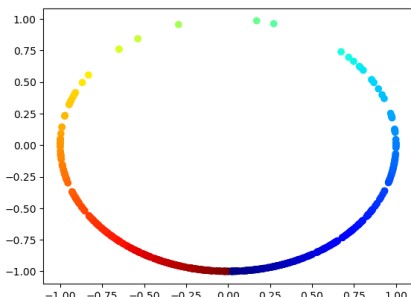



Figure 4: The circular dataset, not uniformly distributed, projected onto the $(x, y)$ plane.

Figure 5: A visual representation of the function values of the estimated invariant function of $X$ as given in 21.

bearing striking resemblance to the ground-truth expression $x^2 + y^2 = 0$.

In the search for infinitesimal generators which annihilate both 19 and 20 in accordance with 13, we perform four (relatively fast) searches: first, among vector fields with constant components; second among vector fields with linear components (no constant terms); third, among vector fields with affine components; fourth among vector fields with quadratic components. In this case, since the effective dimension of the dataset is 1, we limit our searches to those where the number of columns of $W$ is 1. Our best result is obtained in the second case, and the estimated vector field is

$$X = (-0.0066x + 0.7047y)\partial_x + (-0.7095x + 0.0038y)\partial_y, \tag{21}$$

which is similar to the expected outcome of $-y\partial_x + x\partial_y$. Lastly, search for a function $f$ such that $X(f) = 0$, assuming $f$ can be approximated by a quadratic polynomial (with no constant term). Our estimate is given by

$$f = -0.0088x - 0.0886y + 0.7392x^2 - 0.0185xy + 0.6673y^2, \tag{22}$$

to which the reader can compare with $x^2 + y^2$. A plot of the function values of $f$ is given in 5.

It appears that Figure 5 is more faithful to the ground-truth invariant function than 3. We do not believe this indicates the inferiority of symmetry detection using probability density estimation. Rather, we believe this may have been caused by the estimation of gradients of the probability distribution: the gradients of the level set functions were computed exactly. This may indicate that a parametric approach to density estimation may allow for better approximation of gradients.

### D.3 Discrete Rotation

This experiment is motivated by the simulated discrete rotation experiment given in [5]. Conveniently, code is provided to reproduce the results of their experiment, which entails the generation of $20,000$ points $(x, y, z)$ resulting in a multivariate normal distribution centered at the origin. Then, target values for a hypothetical regression task are generated via the following function for a fixed integer $k$:

$$f(x, y, z) = \frac{z}{1 + \arctan(x/y \bmod 2\pi/k)}. \tag{23}$$

The task with this dataset is to correctly predict the angle corresponding to the alleged discrete symmetry: an integer multiple of $2\pi/k$. In [5], $k = 7$ is chosen, so we will do the same. Our methods require that we estimate $f$ from the data, then a parametric description of the symmetry group, after which one can estimate the parameters to approximate a suitable group element. However, the function given is difficult to estimate via regression. Thus, we will estimate the density of the dataset using kernel density estimation, using the function values as weights. As kernel density estimation is known to perform poorly in more than two dimensions, we will restrict the size of the experiment by setting $z = 1$.

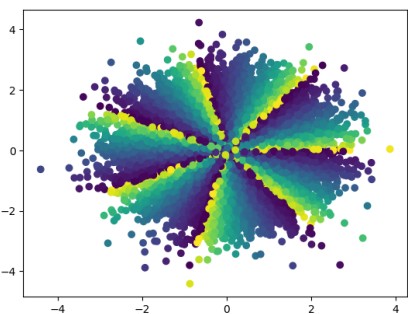

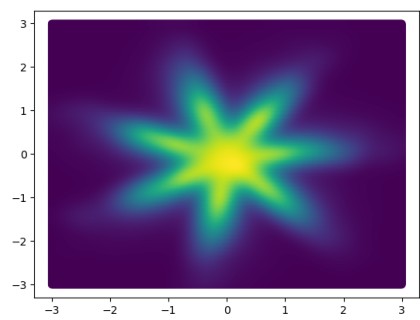

Figure 6: Pictured are the 20,000 points generated by replicating an experiment of [5] restricted to two dimensions.

Figure 7: A visualization of the density estimated using weighted kernel density estimation.

Though we mentioned using the function values as weights for the density estimation, we actually apply $f \to f^8$ (element-wise), so that the differences in weights are more exaggerated. This changes the function to be estimated, but the symmetries of $f$ and $f^8$ are thought to be similar.[6]

We plot the data in Figure 6, and Figure 7 gives a visualization of the weighted kernel density estimation. Notably, the density estimation has "smoothed over" the jump discontinuities of the original function.

With the density $p(x, y)$ estimated, we now must parameterize the transformation group. In this experiment, we will search only among the planar rotations, so that the group can be parameterized by the angle $\theta$:

$$S(\theta) = \begin{bmatrix} \cos(\theta) & \sin(\theta) \\ -\sin(\theta) & \cos(\theta) \end{bmatrix}. \tag{24}$$

We thus minimize the following function $L$ for the parameter $\theta$, subject to $\theta > \pi/6$:

$$L(\theta) = |P(S(\theta)\mathbf{X}) - P(\mathbf{X})|, \tag{25}$$

where $P$ is the vectorization of $p$. We choose $\theta > \pi/6$ so as to find transformations away from the trivial $\theta = 0$. Our optimization, performed using python's *scipy* package, yields a result of $\theta \approx 0.8972$, which is $\frac{2\pi}{7}$ to four decimal places, and our similarity to ground truth is $1.0000$. The total computation time, including the estimation of the density, is approximately 48 seconds.

We repeat the experiment using only 1000 data points, recovering $\theta \approx 0.8333$, in approximately 0.2 seconds. This yields a similarity to ground truth of $0.9979$. On the other hand, when the amount of data for the experiment with LieGAN is moderately reduced, the model performance suffers greatly. The intended generator is

$$\begin{bmatrix} 0 & 1 & 0 \\ -1 & 0 & 0 \\ 0 & 0 & 1 \end{bmatrix}, \tag{26}$$

while the obtained generator is

$$\begin{bmatrix} -0.0869 & -0.0244 & 0.1371 \\ 0.2666 & -0.4993 & -0.4303 \\ 0.0841 & 0.1345 & 0.0377 \end{bmatrix}. \tag{27}$$

The similarity of these two generators is $0.1956$. Thus, our approach seems to have greater potential to handle discrete symmetry with smaller datasets in two dimensions, in addition to having a friendlier interface.

---

[6]In fact, identifying the symmetries of a probability distribution $p(\mathbf{x})$ using our methods is equivalent to identifying the symmetries of $p(\mathbf{x})h(\mathbf{x})$, provided that a function $h$ can be given such that whenever $X(p) = 0$, $X(h) = 0$.

### D.4 Infinitesimal Isometries

We will now conduct a controlled experiment to detect symmetries beyond the complication obtainable by current state of the art methods. Specifically, we will detect infinitesimal isometries: that is, Killing vectors. We set up the experiment by creating an artificial data set $\tilde{\mathcal{D}}$ in $\mathbb{R}^3$: we name the coordinates $u$, $v$, and $w$. This data set is created by drawing three random samples of size $2^{12}$ (one sample for each of the three coordinates) from $U(-1, 1)$. Then, we create a data set $\mathcal{D}$ with four features, namely $x$, $y$, $z$, and $t$, by applying the following function for each data point in $\tilde{\mathcal{D}}$:

$$x = u, \qquad y = v, \qquad z = u^2 + v^2 - w, \qquad t = 2u. \tag{28}$$

Next, we generate a list of function values, one for each point in $\tilde{\mathcal{D}}$, according to $f = 9u^2 + v^2 + w$: the goal of the experiment is to recover infinitesimal isometries of $f$.

The experiment begins under the assumption that the starting data set is $\mathcal{D}$, and that the data set $\tilde{\mathcal{D}}$ has been produced from $\mathcal{D}$ by means of a dimensionality reduction algorithm. The computation begins by learning functions $x(u, v, w)$, $y(u, v, w)$, $z(u, v, w)$, and $t(u, v, w)$ with polynomial regression. This we do, with results varying only, it seems, due to numerical approximation error, so that we take 28 to be our estimation.

We will need to compute the Killing vectors for the 3-dimensional space, for which we will need the Riemannian metric. We assume that the metric in the original 4-dimensional space is the Euclidean metric, and we use 28 to pull back this Euclidean metric. The result can be expressed as the following matrix in the coordinates $(u, v, w)$:

$$g = \begin{bmatrix} 4u^2 + 5 & 4uv & -2u \\ 4uv & 4v^2 + 1 & -2v \\ -2u & -2v & 1 \end{bmatrix}. \tag{29}$$

We have computed this metric using Maple [22], even though the geomstats package for python [21] appears to be able to perform this, primarily because geomstats does not appear to currently support the calculation of Killing vectors. We calculate a basis for the Killing vectors in Maple, and we note the six basis elements to be

$$X_1 = \left(u^2 + v^2 - w\right) \partial_u + \left(2u^3 + 2uv^2 - 2uw + 5u\right) \partial_w, \tag{30}$$

$$X_2 = \left(u^2 + v^2 - w\right) \partial_v + \left(2u^2 v + 2v^3 - 2vw + v\right) \partial_w, \tag{31}$$

$$X_3 = \partial_w, \qquad X_4 = -v\partial_u + 5u\partial_v + 8vu\partial_w, \qquad X_5 = \partial_v + 2v\partial_w, \qquad X_6 = \partial_u + 2u\partial_w. \tag{32}$$

We then return to python to learn $f$ via polynomial regression. The result of this regression task is almost exactly $f = 9u^2 + v^2 + w$, so that we take this as our function for which we will seek to estimate symmetries. To identify the symmetries, we perform the following constrained optimization task:

$$\min_{a_i} \sum_{i=1}^{6} a_i X_i(f), \qquad \sum_{u=1}^{6} a_i^2 = 1. \tag{33}$$

This is equivalent to the formulation given in 13 with the feature matrix being defined by the components of the known Killing vectors and with $W$ having six rows and a single column. We handle this particular task using McTorch [16], which is a manifold optimization library compatible with pytorch. The result is a similarity score of 1.0000 with respect to the ground truth of

$$a_1 = a_2 = a_3 = a_5 = a_6 = 0, \qquad a_4 = 1. \tag{34}$$

Thus, our methods have discovered the correct infinitesimal isometry of $f$ in the coordinates $(u, v, w)$.

### D.5 10-Dimensional Reduction using Level Set Estimation

We will now conduct an experiment for a dataset with 10 features. The goal of this experiment is to apply level set estimation to reproduce the correct number and complexity of components of the ground truth level set function. First, we sample $N = 2^{16}$ numbers $\{t_i\}_{i=1}^{N}$ from a uniform distribution on $[-2, 2]$. Then, three additional independent samples are made of the same size: $\{x_i\}_{i=1}^{N}$, $\{y_i\}_{i=1}^{N}$, and $\{z_i\}_{i=1}^{N}$. Thus, the dataset $\tilde{\mathcal{D}} = \{(t_i, x_i, y_i, z_i)\}_{i=1}^{N}$ is a hypercube of dimension 4. Then, a 10-feature dataset $\mathcal{D}$ is constructed as follows for each point $(t, x, y, z) \in \tilde{\mathcal{D}}$:

$$x_1 = t, \quad x_2 = x, \quad x_3 = y, \quad x_4 = z, \quad x_5 = 2t, \tag{35}$$

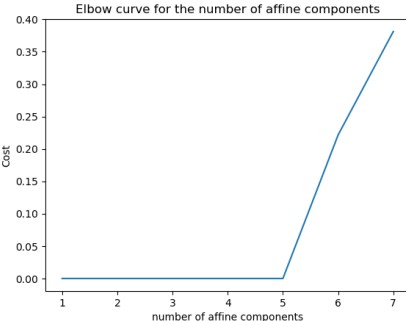

Figure 8: An elbow curve for selecting the number of affine components for the 10-dimensional simulation.

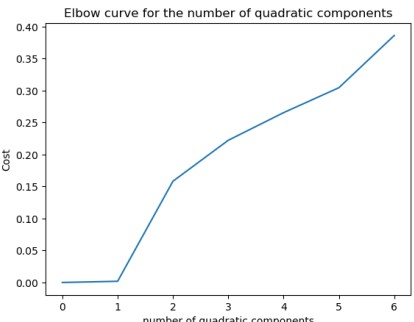

Figure 9: An elbow curve for selecting the number of quadratic components for the 10-dimensional simulation. A value of $0$ is placed at $0$ on the horizontal axis for easier visual comparison of the relatively small value obtained by selecting one component.

$$x_6 = x^2 + y^2 - t, \quad x_7 = 4, \quad x_8 = 0, \quad x_9 = t - z, \quad x_{10} = 1.$$

The equations given in 35 could be used to define a function $\Theta : \mathbb{R}^4 \to \mathbb{R}^{10}$, allowing a Riemannian metric tensor to be given on $\mathbb{R}^4$ by pulling back the Euclidean metric tensor on $\mathbb{R}^{10}$, much like what we demonstrated in a previous example. However, our goal here is level set estimation. Our dataset, by construction, admits two different and equally valid level set descriptions. The first is given by the following equations:

$$x_5 - 2x_1 = 0, \quad x_6 - x_2^2 - x_3^2 + x_1 = 0, \quad x_7 - 4 = 0, \quad x_8 = 0, \quad x_9 - x_1 + x_3 = 0, \quad x_{10} - 1 = 0. \tag{36}$$

Five of the equations given above define affine functions. If this was known from the data, one could project the data onto the hyperplane defined by these equations, obtaining, in coordinates $(x_1, x_2, x_3, x_4, x_6)$, the level set description of

$$x_6 - x_2^2 - x_3^2 + x_1 = 0. \tag{37}$$

We begin applying level set estimation by assuming that all component functions are affine. Beginning by identifying approximate affine components is recommended in an effort to avoid obtaining degenerate equations when expanding the search space to quadratic components. In any case, we iteratively optimize according to 10, constraining $W$ to be an orthogonal matrix, starting the number of columns of $W$ at 1, then increasing by steps of 1 to the number 7. Our optimization routine uses the MSE loss and (Riemannian) gradient descent with a learning rate of $0.01$. A plot of the cost function values as a function of the number of columns of $W$ is given in Figure 8. From this figure, it is evident that the elbow occurs at 5 columns.

Again, to avoid degenerate components, we use the affine components discovered in the previous step to project onto the associated 5-dimensional plane. On the new 5-dimensional space, we apply level set estimation, looking now for quadratic components. This optimization routine uses the L1 loss function and (Riemannian) Adagrad optimizer with a learning rate of $0.01$. A plot of the number of columns of $W$ is given in Figure 9. It seems to be most reasonable to select 1 as the number of approximate quadratic components. The single quadratic component defines a level set estimate of the reduced data space.

### D.6 Invariant Features of a Model-Induced Class Probability Density

In a classification setting, certain models can output class probability predictions for given points. If the distribution function is differentiable, we may also seek vector fields which annihilate this function, allowing us to give the symmetries. In this experiment, we analyze the symmetries of a model-induced probability distribution arising from the Palmer Penguins dataset [24]. This dataset consists of three distinct classes of penguin species with various features, with the total number of

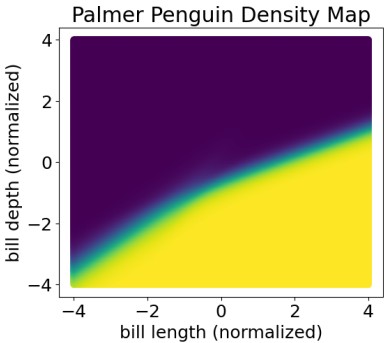

Palmer Penguin Density Map

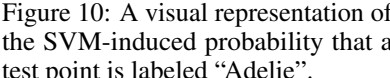

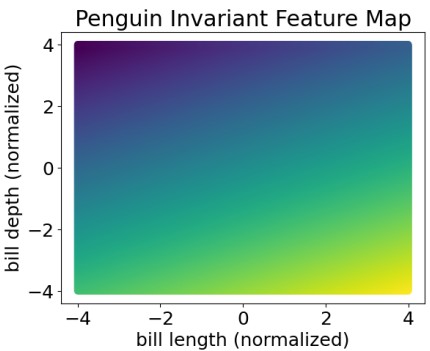

Penguin Invariant Feature Map

Figure 10: A visual representation of the SVM-induced probability that a test point is labeled "Adelie".

Figure 11: A representation of the estimated invariant function of the symmetry of the SVM-induced distribution.

observations being 344. We retain the features called "bill length" and "bill depth," and we train a support vector machine using *scikit-learn* [25]. We take as our probability density the model-induced probability that a test point will be predicted as belonging to class "Adelie." For reference, $80\%$ of the data was used to train the model, and the accuracy of the model when applied to the test set was $0.94$.

A plot of the probability density is given in Figure 10. From the probability density, we estimate a vector field with quadratic components for which the density is invariant. Then, we use the estimated vector field to learn an approximate polynomial invariant function. A plot of the estimated invariant function is given in Figure 11. Visually, the contour lines of the probability distribution approximately align with contour lines of the estimated invariant function, which function is estimated as $0.4335x - 0.9011y$. The feature defined by this estimated function could be used in subsequent machine learning tasks. Suppose, for example, that we encountered additional data points with the task of classifying whether the points belonged to the class "Adelie." One could reduce the dimension of the data to 1, with the single feature being given by the approximate invariant function, which feature would be highly correlated with the probability function for that class.

### D.7 Symmetry-Assisted Regression

The symmetries of a distribution or shape on which data lies may correspond with another machine learning function such as target values in a regression problem. In this experiment, we will show that symmetry-based feature engineering can improve model performance on a regression task. The data for this experiment has been analyzed previously for the purpose of discovering discrete rotational symmetry [5], though we set $z = 1$ for the purpose of demonstrating symmetry-informed feature engineering. We also search for discrete rotational symmetry for this dataset in Appendix D.3.

We generate 2000 data points from a 2-dimensional standard normal distribution. For each data point, function values are generated according to

$$f = \frac{1}{1 + \arctan(y/x \bmod 2\pi/7)}. \tag{38}$$

Using *scikit-learn* [25], we train a random forest regressor on this data using a random train/test split with the test size proportion being $0.2$. The $R^2$ scores for this experiment are given in Table 2.

Without consideration of targets, the dataset is approximately rotationally symmetric. We apply kernel density estimation on the unlabeled data, then test for symmetry of the estimated distribution. The estimated infinitesimal generator is given by

$$(0.015 + 0.0069x - 0.6867y)\partial_x + (-0.0012 + 0.7267x + 0.0080y)\partial_y. \tag{39}$$

We estimate the invariant feature for this infinitesimal generator using our method. The flow parameter is approximately the polar angle, since the vector field given in Eq. 39 has a similarity of $0.9994$ to the vector field corresponding to rotation in the plane about the origin. We thus transform the original dataset to coordinates $(\xi, \theta)$, where $\xi$ is the estimated invariant feature and where $\theta$ is the

Table 2: $R^2$ Scores for Symmetry-Enhanced Regression with Random Forests

| Dataset | $R^2$ score (Train) | $R^2$ score (Test) |
|---|---|---|
| Original | 0.9488 | 0.5965 |
| Transformed | 0.9975 | 0.9965 |

polar angle. We train a random forest model using the transformed data, and our results are given in Table 2. Based on the obtained $R^2$ scores in table 2, it appears that the regression task is more easily accomplished in the symmetry-based feature space than in the original feature space.

# E  A Note about the Freedom Present in Identifying Vector Fields

Given a vector field $X$ and a vector field $fX$, a function $h$ is $X$-invariant if and only if it is $fX$-invariant. If $h$ is $X$-invariant, the flow of $X$ is a symmetry of $h$, so that the flow of $fX$ is also a symmetry of $h$: this follows from our discussion of vector fields in section 2.

For example, a circle centered at the origin, characterized by $F = 0$ where $F(x, y) = x^2 + y^2 - r^2$ for some real number $r$, exhibits rotational symmetry described by $X = -y\partial_x + x\partial_y$. However, $\frac{1}{y}X(F) = 0$, so that the flow of the vector field $-\partial_x + \frac{x}{y}\partial_y$, where defined, is also a symmetry of the circle.

This may seem to introduce a theoretical problem requiring greater care when detecting non-affine symmetry. However, our proposed method of constructing models which are invariant with respect to the symmetry group requires only the identification of functions which are invariant with respect to the vector fields. Thus, both $X$ and $fX$ are valid answers for "ground truth" symmetries, since a function $h$ is $X$-invariant if and only if it is $fX$-invariant.

We note that this lack of uniqueness does present a challenge when estimating suitable vector fields. When estimating the symmetries via constrained regression, $X$ and $fX$ may or may not both be present in the search space. As mentioned in section 3, one potential workaround is owing to the fact that symmetry estimation can be done symbolically. Consider the example of $F = 0$ given previously in this section. The Jacobian matrix of $F$ is given as

$$J = \begin{bmatrix} 2x & 2y & 2z \end{bmatrix},$$

and a basis for the nullspace of $J$ (using functions as scalars) is

$$\left\{ \begin{bmatrix} -y \\ x \\ 0 \end{bmatrix}, \begin{bmatrix} 0 \\ -z \\ y \end{bmatrix} \right\},$$

which vectors correspond to tangent vectors $-y\partial_x + x\partial_y$ and $-z\partial_y + y\partial_z$, respectively. The flows of these vector fields correspond with rotations (about the origin) in the $(x, y)$ and $(y, z)$ planes, respectively. However, only two vector fields can be recovered using this method, which in this case has neglected another symmetry, namely $-z\partial_x + x\partial_z$, corresponding to rotations in the $(x, z)$ plane. In fact, each of the three vector fields corresponding to rotations about a coordinate axis are isometries of the sphere described by $F = 0$. We should note, however, that the generator for rotations in the $(x, z)$ plane can be expressed as a linear combination of the other rotations using functions as scalars:

$$\frac{z}{y}\left(-y\partial_x + x\partial_y\right) + \frac{x}{y}\left(-z\partial_y + y\partial_z\right) = -z\partial_x + x\partial_z.$$

Therefore, it is plausible that applications exist in which a symbolic approach to continuous symmetry discovery may be sufficient. Our approach does not rely on symbolic software, however.

We also mention that the issue of unwanted freedom can be addressed by identifying "special" vector fields and reducing the search space of vector fields to linear combinations of these special vector fields known as Killing vectors. This would eliminate the uniqueness problem and is experimented with in Appendix D.4. This approach is a special case of symmetry detection where only isometries can be detected.

Another point which may help to address concerns about our handling of the difficult nature of non-affine symmetry detection is in the relationship between the flows of $X$ and $fX$ generally. The

trace of the flow of $X$ through the point $p$ is characterized by the level set $h_i = c_i$, where $\{h_i\}$ is a complete set of $X$-invariant functions. Since a function is $X$-invariant if and only if it is $fX$-invariant, a complete set of invariant functions for $fX$ can be taken to be $\{h_i\}$ without loss of generality. Thus, the level set $h_i = c_i$ also characterizes the trace of the flow of $fX$ through the point $p$. This argument assumes that the flows of $X$ and $fX$, as well as the vector fields themselves, are well-defined in an open neighborhood about the point $p$.

In fact, in our example above with $F = x^2 + y^2 - r^2$, the flow of $-\partial_x + \frac{x}{y}\partial_y$, assuming $y > 0$, is given as

$$\Phi(t, (x, y)) = \left(x - t, \sqrt{y^2 + 2tx - t^2}\right),$$

which trace is a (part of a) circle (we note that the flow parameter is $-x$). The trace of this flow through a point is equivalent to the trace of $X$ through the same point.

