# OpenReview forum: "Symmetry Discovery Beyond Affine Transformations"
_NeurIPS.cc/2024/Conference — NeurIPS 2024 poster_

### Official Review · Reviewer_U1Sk · 2024-07-08

**Soundness:** 2
**Presentation:** 2
**Contribution:** 2
**Rating:** 4
**Confidence:** 4

**Summary:**

This paper proposes a method for finding a transformation that is invariant to a given function. The transformation is restricted as governed by a single parameter, so it is written as a flow described by a specific vector field. The method estimates the vector field by solving the polynomial regression. The authors evaluated the performance on three toy datasets and a classification dataset.

**Strengths:**

1. A flow-based symmetry detection is novel and an interesting research direction.

**Weaknesses:**

1. There is a large room to improve the presentation. First of all, the terminology seems not standard in ML (e.g. I've never heard the term "machine learning function"). Second, the contents are not self-contained --- some unfamiliar notions appear without explanation (e.g. elbow curve). Third, it is not clear what the proposed method actually does. For example, it is said that the method solves Eq.(10), but it is not likely that we can always solve it. I mean, for some data, the equation may not have a solution. Such a case is not mentioned in the paper.
2. The capabilities and limitations of the proposed method are not fully mentioned.
3. The experiments are mainly on toy datasets and are not strongly convincing that the method is applicable to real problems.

**Questions:**

1. What is the transformation class that is (theoretically) handled by the method?
1. How did you solve Eq. (10), (13), (14) in the experiments?
1. What would happen if x contains some noise?
1. When does the proposed method fail?

**Limitations:**

Limitations are not provided.

---

> ### Author Rebuttal · Authors · 2024-08-07
>
> We thank the reviewer for their comments. We have responded below.
>
> **Terminology**: With no intent to confuse the reader, we used the term "machine learning function" to cover very general types of functions which may appear within the context of machine learning. Such a function could be a regression function, a classification function, or a function offering a manifold description of given data, as in level set estimation or in the presence of a metric tensor.
>
> The elbow curve notion refers to a relatively sudden increase in function values, perceived as an elbow shape, since the function values immediately preceding the elbow point are usually close together. This is sometimes referred to as finding a "kink," as in [R1] within the context of selecting the "best" number of clusters $k$ for $k$-means clustering.
>
> In general, Eq. (10) may not have a solution, as is the case with OLS regression. Eq. (10) is estimated/fitted by constrained optimization of a selected loss function. We will clarify this in the revision.
>
> **Capabilities and Limitations**: We have clarified many points regarding the capabilities and limitations of our method in our response to the reviewers. We will include a discussion of these points in the revised version.
>
> **Only using toy datasets**: Current SOTA papers in symmetry detection commonly use simulated data. An experiment with simulated data allows for the identification of ground truth symmetry, allowing one to quantify the accuracy of a given method. Symmetry detection on real-world data can be applied, but it is harder to evaluate the ability of new methods to properly detect symmetry if ground truth symmetry is not known, as is generally the case with real data. Thus our experiments are consistent with other SOTA papers in this area.
>
> Nevertheless, we present an additional experiment on real data that we will include in the revision. This comes from a publicly-available dataset that deals with weather for four decades in the vicinity of Bear Lake, Utah. The dataset, along with a report describing how the data was sourced, is publicly available [R2].
>
> The dataset gives daily weather attributes. It contains $14,610$ entries with 81 numeric attributes including the daily elevation level of the lake. The dataset contains precisely 40 years' worth of data from October of 1981 through September of 2021.
>
> We believe an understanding of the behavior of the weather in the past is relevant to this problem. Therefore, we first construct $13,149$ time series of length $1461$ in $81$ dimensions by means of a sliding window of length $1461$ days: the first time series is constructed using the first $1461$ days (the number of days in four years). The next time series is constructed using the second day through day 1462, and so forth.
>
> After converting the raw data to time series data, we apply a transformation on the data meant to extract time-relevant features of the data  known as the Multirocket transform [R3]. We select $168$ kernels in our transform. The Multirocket transform transforms the data from $13,149$ time series of length $1461$ in $81$ variables to tabular data: $13,149$ entries in $1344$ variables.
>
> For such high-dimensional data, we turn to PHATE, a SOTA data visualization method [R4]. Using PHATE, we reduce the dimension to $2$, so that our new dataset has $13,149$ entries in $2$ variables. The resulting data appears to approximately lie on a circular shape and is shown in Figure 1 of the rebuttal attachment.
>
> Figure 1 in the rebuttal attachment suggests that the data is largely periodic. In fact, further experimentation reveals that the points for a given calendar year correspond to a traversal around the circular-like shape. Thus, for the analysis of non-seasonal weather patterns, it may be of use to examine features of this embedded dataset which are invariant under the periodic transformation, the approximate symmetry.
>
> Indeed, our method reveals an approximate invariant function given by
>
> \begin{equation*}
>     f(x,y) = 0.33592 x^2 + 0.94189 y^2 - 2.9743 \cdot 10^{-4} = 0.
> \end{equation*}
>
> In Figure 2 of the rebuttal attachment, we replot the embedded data, colored this time by the value of the approximate invariant function. This experiment shows that symmetry can occur in real data, and that our method can detect symmetry and estimate invariant functions for real data.
>
> **Transformation class**: The transformations our method can handle are 1-parameter subgroups of transformation groups (Lie groups). We will clarify this in the revision.
>
> **Solving method**: Solutions to Eqs. (10), (13), and (14) are estimated using constrained regression. The specific loss function and hyperparameters vary by experiment, but we will clarify these details in the revision.
>
> **Noise**: Noise is common in regression problems, and it affects the quality of least squares estimates. We have noise present in the experiment that compares our method with LieGAN, since the dataset is not technically rotationally symmetric, but rather approximately rotationally symmetric.
>
> **When our method fails**: It may fail if the number of independent parameters exceeds the number of datapoints. The success of the method depends on optimization and loss function choices, as well as hyperparameters. We will clarify these points in the revision.
>
> [R1] T. Hastie, R. Tibshirani, and J. Feiedman, ``The Elements of Statistical Learning: Data Mining, Inference, and Prediction,'' 2nd ed. Springer, 2009.
>
> [R2] B. D. Shaw et al., “Supplementary files for: ‘interactive modeling of bear lake elevations in a future climate’,” 2024.
>
> [R3] C. W. Tan, A. Dempster, C. Bergmeir, and G. I. Webb, “Multirocket: Effective summary statistics for convolutional outputs in time series classification,” CoRR, abs/2102.00457, 2021.
>
> [R4] K. Moon et al., “Visualizing structure and transitions in high-dimensional biological data,” Nature Biotechnology, vol. 37, pp. 1482 – 1492, 2019.

---

> > ### Comment · Reviewer_U1Sk · 2024-08-13
> > **To authors**
> >
> > Thank you for responding to my review. I also appreciate that the authors conducted additional experiments. My concerns are partially resolved, and I will increase my score.

---

### Official Review · Reviewer_uBWo · 2024-07-08

**Soundness:** 3
**Presentation:** 4
**Contribution:** 2
**Rating:** 6
**Confidence:** 4

**Summary:**

The paper presents a method for continuous symmetry detection under the manifold assumption.
Crucially, the symmetries that are discovered by this method can extend beyond the affine ones.
The method is tested and compared against the state of the art (LieGAN), and is found to outperform it in scarce data regime, and be competitive in the large data regime.

**Strengths:**

* The paper connects symmetry to invariance of dynamical systems to particular functions. Instead of explicit identification of the symmetry, the focus is on identifying the infinitesimal generators of the flows which correspond to the symmetries themselves. Once found, the vector field can be used to create symmetry-invariant features.
* The exposition is overall very clear. Many examples are provided throughout.
* The paper outlines a (sometimes) computationally cheap alternative to methods such as LieGAN.

**Weaknesses:**

* Depending on the machine learning function of interest, the method requires level set estimation, which is done by assuming that level sets themselves can be found by linear combination of polynomials. This seems to be very optimistic
* The elbow method to identify the right amount of polynomial terms to keep seems a bit brittle
* The scaling of a polynomial based method is very unfavourable beyond very toy-like examples.
* More generally, who chooses the feature functions that form the basis for the linear combinations?

If these weaknesses are addressed, the rating from this reviewer may increase.

**Questions:**

Can we get a discussion of the computational complexity/scaling of this method? Seems like there are combinatorial explosions hiding just beneath the surface of the method. A proper discussion of this may result in an increase in the rating from this reviewer.

**Limitations:**

The authors have properly discussed the limitations of their model in terms of the experiments they have run. Not enough has been said in terms of the various null space estimations, which require knowledge of adequate “basis functions”, which can not just be polynomials in most cases.

---

> ### Author Rebuttal · Authors · 2024-08-07
>
> We thank the reviewer for their comments. We have responded below to the concerns raised.
>
> **Use of Polynomials**: In general, level set estimation can be applied using linear combinations of any set of smooth functions, though our experiments do specialize to polynomial functions. However, smooth functions can be approximated by polynomials of sufficiently high degree and are thus universal approximators [R1].
>
> **The elbow curve**: Instability can arise, and the results of the elbow curve can be misleading. This seems to primarily occur when the problem of degenerate expressions, discussed first in lines 182 - 204, is not properly controlled. For level set estimation, if both $f=0$ and $hf=0$ are discoverable expressions based on the pre-defined search space (for example, $x-y=0$ and $x^2-xy=0$), the loss function values may not generally follow an elbow curve, and any perceived elbow may not occur at the correct number of components for level set estimation. We discuss some workarounds for this problem, both for level set estimation and for vector field estimation, in lines 182-204, as well as 226-231.
>
> **Scaling beyond toy examples**: Vector fields with polynomial coefficients can characterize highly complex and non-trivial symmetries. Especially due to their universal approximation properties [R1], we expect that many symmetries that occur in the real world are expressible or approximately expressible in this form. Additionally, our method advances the current SOTA methods, as current SOTA methods consider a more restricted class of symmetries. Thus, we believe our contribution is an important advance. See our overall rebuttal for more details on our contributions. See also our response to reviewer U1Sk for an experiment on real data.
>
> **Feature function choice**: The features are chosen, in general, based on the types of symmetries sought. For example, if only affine symmetries are being sought, the feature functions for the vector fields are chosen to be affine functions. If no assumption on the form of the symmetries can be made, one can choose the features to be arbitrary polynomials, due to their universal approximation property [R1].
>
> **Computational complexity**: The computational complexity of polynomial regression is comparatively low, especially in a single dimension [R2]. Moreover, manifold optimization algorithms on the matrix manifolds used herein are also of low computational complexity, with a recent manifold optimization algorithm obtaining computational complexity of $\mathcal{O}(\log(T)/\sqrt{T})$, where $T$ is the number of iterations [R3]. In contrast, current SOTA methods employ large neural networks, which networks are known to take more computational resources to train.
>
> There is no combinatorial explosion in our method. In fact, we can obtain a precise count of the number of parameters our method requires. Each vector field in $m$ dimensions using polynomials of degree $n$ has the following number of coefficients:
>
> \begin{equation*}
> \dfrac{(m+n)!}{n!(m-1)!}.
> \end{equation*}
>
> The coefficients can increase quickly, particularly in higher dimensions, but for every fixed value of $m$, the number of coefficients increases as a polynomial function of $n$ rather than combinatorially. However, each coefficient in our model increases the power of our method to detect symmetries. In contrast, while SOTA methods employing large neural networks must have parameters like ours which characterize the symmetries, they also have additional parameters. The number of parameters in our method increases according to the number of vector fields in a basis of our search space, whereas in methods such as LieGAN, these parameters are required in addition to other parameters present in the neural network architecture.
>
> To address the potential issue of having a “large” number of parameters, which number is dwarfed by the number of parameters present in neural networks that take $m$ inputs, we suggest limiting the search space to infinitesimal isometries: Killing vectors, that is. We discuss this restriction further in Appendix A.
>
>
> [R1] A. Pinkus, “Weierstrass and approximation theory,” Journal of Approximation Theory, vol. 107, no. 1, pp. 1–66, 2000.
>
> [R2] L. Li, “A new complexity bound for the least-squares problem,” Computers \& Mathematics with Applications, vol. 31, no. 12, pp. 15–16, 1996.
>
> [R3] H. Kasai, P. Jawanpuria, and B. Mishra, “Riemannian adaptive stochastic gradient algorithms on matrix manifolds,” 2019.

---

> > ### Comment · Reviewer_uBWo · 2024-08-08
> >
> > I wish to thank the authors for their careful and thorough addressing of this reviewer’s questions.
> > The discussion of the scaling in terms of ambient dimension and polynomial degree is well received; while there may not be a combinatorial explosion for fixed ambient dimension as a function of the chosen degree, high ambient dimension still seems to be a non-ideal setting.
> > In a related way, the example provided by the authors on a real world dataset seems to have been conducted on a projection of the original, high dimensional dataset (via application of PHATE). While such a procedure is certainly appropriate and convincing in this case, it is not clear if it is also necessary for the success of the proposed method of symmetry discovery.
> > After considering the answers currently provided in this rebuttal, this reviewer confirms their rating.

---

> > > ### Author Response · Authors · 2024-08-10
> > > **Extension of experiment to high dimensional data**
> > >
> > > We thank the reviewer for the feedback. In light of the question of conducting the method in a high dimensional setting, we have applied our method directly to the rocket-transformed Bear lake data of dimension 1344. We provide a summary of the experiment herein.
> > >
> > > We begin level set estimation by restricting our search space to polynomials of degree 1 (or less). This we do to avoid the discovery of degenerate components of the level set function. The first elbow curve is obtained using increments of integer multiples of 84, so that each subsequent iteration corresponds to an integer multiple of 84 components of the level set model. We identify the number $15 \cdot 84$ as the elbow point. For having $15 \cdot 84$ components of degree 1 polynomials, we find that the fitting is completed in approximately $555$ seconds. We have used the MSE loss and Riemannian SGD with a learning rate of $0.01$, training for $1000$ epochs. (We recall that, for the sake of comparing SOTA methods, LieGAN required approximately 175 seconds to detect symmetry in dimension 2, training for 100 epochs.)
> > >
> > > Having obtained a degree 1 polynomial estimation of the data, we project the data of dimension 1344 onto the space of dimension $1344-15 \cdot 84 = 84$, similar to what we have done in our experiment in 10 dimensions in appendix C5 the paper itself. Realizing there may still be degree 1 polynomial terms, we generate another elbow curve, this time in incremental steps of 12. We find an elbow point at $6 \cdot 12$, training with the same loss function, number of epochs, and optimization parameters and algorithm as before. These iterations take significantly less time, being in a lower number of dimensions, with the iteration at $6 \cdot 12$ components taking approximately 3 seconds. Again, we project the data onto the lower dimensional space implied by our level set description of the data. (By the way, this projection is, in effect, using constant-polynomial vector fields to obtain manifold coordinates as the flow of these vector fields. This is possible since the vector fields are so simple. We do not explicitly identify the vector fields: see the 10-dimensional experiment in appendix C5 for an analogous experiment.)
> > >
> > > We continue the search for degree 1 polynomial components of the level set function, now in a mere 12 dimensions. Another elbow curve reveals an elbow point at 8, and we again project the data so that the final dataset exists in 4 dimensions. Now that we are confident that no additional degree 1 polynomial terms exist, we expand our search space to polynomials of degree 2. Finding no convincing elbow curve, the level set estimation step concludes, having found no components of complexity greater than degree 1 polynomials in this case.
> > >
> > > The symmetry of the level set function was exploited to explicitly reduce the dimension of the dataset, owing to the simple nature of a level set function with strictly degree 1 polynomials. Therefore, no additional symmetry detection efforts are necessary, and our experiment is concluded.
> > >
> > > This experiment shows that our method can be applied to datasets of higher dimension. Future work includes the study of symmetry for high dimensional datasets. To date, our method appears to be the only method capable of conducting continuous symmetry detection in high dimensions.

---

> > > > ### Comment · Reviewer_uBWo · 2024-08-12
> > > >
> > > > Thank you for this high dim example. To my understanding, there seems to be a focus on the level set estimation step of the method, as opposed to something akin to the invariant circle observed in the PHATE example. This seems to confirm that the overall method is heavily dependent on the level set estimation step, which makes me think that future investigation on this is warranted, especially in high dim setups. I will thus keep the score to 6, and thank the authors for their efforts.

---

> > > > > ### Author Response · Authors · 2024-08-12
> > > > >
> > > > > We thank the reviewer for their feedback. We ask for the consideration of another point on this topic.
> > > > >
> > > > > The operating principle behind our method is the equation $X(f)=0$: we seek vector fields $X$ which annihilate smooth machine learning functions. Thus, our method necessarily relies on tasks which learn a machine learning function $f$. This function can be, but is not necessarily, obtained by means of level set estimation. This occurs before vector fields $X$ can be estimated.
> > > > >
> > > > > Where perhaps disadvantageous at first glance to require first the estimation of a machine learning function before attempting to detect symmetry, we have shown that our method offers a large computational advantage over the current SOTA while comparing with SOTA in terms of accuracy, with the added benefit of being able to detect non-affine symmetries. Additionally, the identification of a specific machine learning function makes symmetry detection well-defined, as it is clear that we are detecting the symmetries of a given function. In contrast, recent methods, including SOTA methods, attempt to detect symmetry in raw data, and it is not clear whether the geometric structure or the distribution of the data is being examined for symmetry.
> > > > >
> > > > > It may also initially appear that the symmetry identified using PHATE is substantially different than the symmetry discovered using level set estimation. However, this is merely a coordinate representation issue. Consider the following example. In coordinates $(u,v)$, suppose the vector field $X = \partial_v$ is taken to be an infinitesimal symmetry of a function $f$. We note that the flow of $X$ is a translation in the $v$ direction. This is analogous to the experiment we did, since the level set estimation approach to the high dimensional experiment yielded only  simple symmetries. Now consider the following transformation $\Phi$ mapping $(u,v)$ coordinates to $(x,y)$ coordinates:
> > > > >
> > > > > \begin{equation*}
> > > > >        \Phi(u,v) = \left(u\cos(v), u\sin(v)\right).
> > > > > \end{equation*}
> > > > > In coordinates $(x,y)$, $X = -y\partial_x + x\partial_y$, which vector field is generally considered to be representative of rotational symmetry. Thus, the way in which $X$ is expressed depends on the coordinates used.
> > > > >
> > > > > Established manifold learning methods such as PHATE introduce new coordinate systems, which subsequently affects the way in which symmetries are expressible. This point is touched on in appendix A, where we question the commonly-accepted assumption that the features of a given dataset should be associated with Euclidean coordinates. Thus, our method is critical in the detection of symmetry, since coordinate transformations (such as PCA) are very common in machine learning. Where the symmetries in one set of coordinates may be considered to be affine, the way in which this symmetry is expressed may substantially change under a simple coordinate transformation. We see this in appendix C4, where the infinitesimal isometries, expressible in terms of affine functions in the original Euclidean coordinates, are expressible in terms of degree 3 polynomials under a simple coordinate transformation.
> > > > >
> > > > > Additionally, given a function and its inverse from the dataset in 1344 dimensions and the PHATE-reduced dataset in 2 dimensions, symmetries in one set of coordinates are expressible in terms of the other coordinates. This relies upon the manifold assumption, which is that the data in 1344 ambient dimensions lies, or is at least well-modeled as lying, on a 2 dimensional manifold described by the PHATE transformation. Explicitly writing vector fields in one set of coordinates in terms of vector fields in other coordinates requires an invertible transformation from one set of coordinates to another, made possible with existing methods such as GRAE [R1].
> > > > >
> > > > > [R1] A. F. Duque, S. Morin, G. Wolf and K. R. Moon, "Geometry Regularized Autoencoders," in IEEE Transactions on Pattern Analysis and Machine Intelligence, vol. 45, no. 6, pp. 7381-7394, 1 June 2023, doi: 10.1109/TPAMI.2022.3222104.

---

### Official Review · Reviewer_fxa1 · 2024-07-13

**Soundness:** 3
**Presentation:** 2
**Contribution:** 2
**Rating:** 5
**Confidence:** 4

**Summary:**

The paper uses standard ideas from differential geometry to find symmetries in datasets.
The procedure is the following:
1. Estimate a parameterization of the dataset (what the authors call machine learning functions). This step looks like manifold learning.
2. Find a vector field under which the machine learning functions are zero.
3. Find a coordinate system for the invariant space for the vector field in 2.

Though the employed techniques are not exactly the same, the ideas remind me of this paper https://arxiv.org/abs/2008.04278

**Strengths:**

- The problem the paper addresses is very interesting and can have many applications.
- Several numerical examples are presented.

**Weaknesses:**

- The paper should state the assumptions under which the algorithms work.
  - In particular it seems that one necessary assumption for the first step is that the data lies in a manifold that can be parameterized by a single chart (the f). Is that correct or can this assumption be bypassed somehow?
  - Another seemingly needed assumption is that the group of symmetries one can learn is a 1-parameter group. If one has a 2-parameter group then it would be given by another vector field, and it is not obvious how to make sure that the 2 vector fields are compatible. Is this a necessary assumption too?

- The paper mentions that the techniques they develop work also for discrete groups, and it gives a not very detailed discussion in Appendix B. I don't see how that's the case. Say that the data has a symmetry with respect to an unknown action of an unknown permutation group. How can this method find it? It is not obvious how the algorithm would work in this case.

**Questions:**

In addition to the questions above:
- What is the rationale behind assuming that the h functions are polynomials? Could this be done by implementing the h functions with MLPs?

**Limitations:**

The paper doesn't explicitly state the technical limitations of their approach (see weaknesses). The paper could be improved significantly by stating the mathematical assumptions under which the algorithms work.

Additional discussions on the dependence on the dimensionality of the data, dimensionality of the symmetry group, number of samples needed, etc, would improve the paper as well. Even if just empirical.

---

> ### Author Rebuttal · Authors · 2024-08-07
>
> We thank the reviewer for their comments. We have responded below to the concerns raised.
>
> **Assumptions**: The single chart assumption is not necessary. Consider $f(x,y,z) = -x^2-y^2+z^2-1$. The surface $f=0$ is a hyperboloid of two sheets. We can estimate a continuous symmetry of this hyperboloid--a rotation in the $(x,y)$ plane--though a single coordinate chart cannot be given. Level set estimation does not deal with coordinate charts, which is a benefit of using a level set to characterize an embedded manifold.
>
> We are also not assuming that the symmetry group has dimension 1. We do assume that the group has 1-parameter subgroups. There may be several 1-parameter subgroups, which our method can handle. The discovered vector fields are compatible in the sense that they admit a common set of invariant functions. This is described in lines 112-120, where we seek vector fields (each representing a 1-parameter subgroup) which annihilate the machine learning functions.
>
> **Discrete transformations**: If one can express the discrete transformation parametrically, this method can be applied to discrete transformations. Many examples stem from continuous symmetries with a fixed continuous parameter, such as a rotation in the plane by a fixed but arbitrary angle $\theta$, where $\theta$ is the parameter to be optimized.
>
> Another example is a 2-d reflection about a straight line passing through the origin. Such a line can be characterized by the equation $ax+by=0$, and a formula for the reflection can be written as
>
> \begin{equation*}
>         S(x,y;a,b) = \dfrac{1}{a^2+b^2} \begin{bmatrix}
>             a^2-b^2 & -2ab\\\\
>             -2ab & b^2-a^2
>         \end{bmatrix}
>         \begin{bmatrix}
>             x\\\\
>             y
>         \end{bmatrix}.
> \end{equation*}
> Optimizing the parameters $a$ and $b$ in $f(S(x)) = f(x)$ for some function $f$ would yield the "best fit" line of reflection under which $f$ is symmetric.
>
> Permutations may not be expressible parametrically, and we will state this explicitly as a limitation of the method, since previous work with permutation groups has been done and is of interest.
>
> **Polynomials vs. MLPs**: Since MLPs are universal approximators (as are polynomial functions), our method can be applied to MLPs as well. The main reason MLPs are not used here is because of the difficulty in using them for level set estimation: the MLP would need to output 0 for every input and yet not be the zero function. Meanwhile, we assume that the invariant functions are polynomials primarily for two reasons. First, in the case of affine symmetry, the invariant functions are typically expressed in terms of polynomials. Secondly, polynomials can approximate smooth functions with an arbitrary degree of accuracy [R1] and are easy and transparent to manipulate.
>
> **Additional Discussion**: We will include a discussion on the number of parameters needed to estimate the symmetries in dimension $m$ using degree $n$ polynomials. This point is discussed in response to reviewer uBWo. The number of parameters relates to the dimension of the symmetry group and can also provide insight on the number of samples needed.
>
> [R1] A. Pinkus, “Weierstrass and approximation theory,” Journal of Approximation Theory, vol. 107, no. 1, pp. 1–66, 2000.

---

> > ### Comment · Reviewer_fxa1 · 2024-08-12
> >
> > I appreciate the response by the authors. The assumption on the 1-parameter subgroups is clear, and I suggest it is stated as an assumption under which the algorithm works more explicitly.
> > The clarification of when one can use this method with discrete groups is useful too. I understand this is not the main point of the paper but since it is mentioned in the abstract I think adding the assumption is necessary.
> > The assumption on the manifold is a bit less clear. Does it work for any manifold or does it need to be the level set of a polynomial or system of polynomials?
> >
> > I'll increase my score but for the final version I'd like to see a remark stating the precise mathematical assumptions on the data an symmetry group which is assumed by the algorithms. it doesn't have to be a theorem.

---

> > > ### Author Response · Authors · 2024-08-13
> > >
> > > We thank the reviewer for their response. Level set estimation works under the assumption that the data lies, or at least approximately lies, on an embedded submanifold of the original feature space. Every embedded submanifold can be represented as a level set of a smooth function, at least locally [R1].
> > >
> > > When a smooth machine learning function $f$ is given, we can estimate vector fields using the equation $X(f)=0$. The function $f$ may be related to level set estimation, or it could be another type of machine learning function, such as a regression function. We assume that the dataset lies on a differentiable manifold $M$, that the function $f$ is a smooth function on $M$, and that $X$ is a tangent vector field on $M$. Though our examples commonly examine symmetries of level sets with polynomial components, it is not necessary to assume that the components of any level set are polynomials.
> > >
> > > We will clarify these assumptions in the revision.
> > >
> > > [R1] J. M. Lee, Introduction to Smooth Manifolds. Springer New York, NY, 2012.

---

### Official Review · Reviewer_S4uK · 2024-07-13

**Soundness:** 3
**Presentation:** 2
**Contribution:** 2
**Rating:** 6
**Confidence:** 3

**Summary:**

The authors are trying to solve a challenging problem to discover symmetry for given data, regarding such symmetry may include non-affine transformations. They observes one-parameter family of symmetric transformations can be represented as a vector field. In the proposed method, they first find machine learning functions (level set estimation) for given data, and then find vector fields which annihilate the machine learning functions.

**Strengths:**

The authors address an interesting problem: symmetry detection beyond affine transformations. Their observation to connect transformations and vector fields technically sound.

**Weaknesses:**

While the title of the manuscript is quite general, their method seems to highly rely on choice of pre-determined models for machine learning functions (Sec. 3.1) and vector fields (Sec. 3.2). However, no experiment addresses impact of choices of those pre-determined models.

In addition, all of the experiments assume we already know a proper pre-determined model of vector fields. The case that pre-determined parametric model of vector fields does not cover the GT vector field used for data generation is not discussed.

To address the challenging problem about non-affine symmetry, we should more carefully handle it. Please see the question about the cosine similarity in "Questions" below.

Results of affine symmetry detection (Sec. 4.1) seem worse than LieGAN. While LieGAN produces results which clearly converge to GT in terms of both bias and variance, but the proposed method does not in terms of both. The proposed method have benefit at low samples and speed, but this contribution seems too marginal. For non-affine symmetry experiment (Sec. 4.2), experimental details are missing. I cannot consider it is a good result unless I can see more case studies about different choices of pre-determined models.

These weaknesses make me doubtful about results and benefit of the proposed method.

**Questions:**

Discuss the following relevant reference:

* Desai, Krish, Benjamin Nachman, and Jesse Thaler. "Symmetry discovery with deep learning." *Physical Review D* 105.9 (2022): 096031.



The cosine similarity metric used in Section 4 is computed by coefficients of the pre-determined model which determine the overall vector field rather than an average of cosine similarities for vectors at each point, right? If so, how about the following case: In Sec. 4.2, even if we got a vector field $2y f \partial_x + 3x^2 f \partial_y$, then it is also a correct vector field describing symmetry of given data. If you are trying to propose a metric to evaluate non-affine symmetry, then your metric should produce zero value also for this case. Please discuss about this circumstance.



Which pre-determined model for vector field are you using in Sec. 4.2? It must be clarified.



The result in Equation (16) seems able to be also produced by LieGAN, since the equation indicate an affine symmetry. Why are not you comparing with LieGAN?



Issues on clarity of exposition.

* Attach color bars for Figures 1-2
* It would be better to clarify to which values $B$ and $w$ in Eq. (1) and $M$ and $W$ in Eq. (13) corresponds for each experiment in Section 4.



Minor typoes

* Line 61: "exampe" -> "example"
* Equation (2): $\partial_{x^i}$ -> $\partial x^i$ at the denominator

**Limitations:**

Following limitations should be emphasized, if my understanding is not wrong.

* Based on pre-determined model of one-parameter symmetries, this work find the most suitable one-parameter symmetry among the pre-determined model

---

> ### Author Rebuttal · Authors · 2024-08-07
>
> We thank the reviewer for their comments. We have responded below to the concerns raised.
>
> **Pre-determined models**: We believe that we are the first to detect continuous symmetries in the context of machine learning beyond the affine group, despite work on the subject spanning decades. Our method of estimating vector fields is regression with general additive models, where the coefficients of the model features are constrained: we specialize to polynomials for the purpose of illustration and better comparison with existing methods for symmetry detection. We have showcased examples in which polynomial-based models are most appropriate, and although experiments which make use of other types of smooth functions can be devised, the only aspect of our method which changes in these cases is the components of the feature matrix used to estimate the coefficients. Therefore, our method can be trivially adapted to non-polynomial models. Later in this response, we will present the results of an experiment which uses non-polynomial functions.
>
> A suitable pre-determined model for the detection of manifold symmetries would rely on the nature of the manifold itself. However, it appears that although the manifold assumption is widely used in machine learning, very little work has been done to characterize the manifolds on which data is assumed to lie. Therefore, our choice to specialize to polynomial-based models is no less valid than other specializations. Additionally, polynomials can be used to approximate any continuous function [R1] and thus, in theory, can characterize any continuous symmetry, at least locally.
>
> **Covering the GT vector field**: If the degrees of the polynomial functions of a vector field are sufficiently high, the vector field can approximate a suitable ground truth vector field. We also reiterate that current SOTA methods, if recharacterized in terms of vector fields, would assume that the coefficients of the vector fields are degree 1 polynomials. Our presentation of symmetry detection in terms of vector fields opens up a great many possibilities for types of functions used, although we use polynomials for better comparison with existing methods and due to their universal approximation property [R1].
>
> **LieGAN comparison**: For every value of $N$, our method usually outperformed LieGAN for most trials. However, the average score of our method was brought down by outlier trial runs for our method, likely due to poor initialization of the model parameters. However, such a shortcoming could be easily overcome by fine-tuning. In the spirit of a fair comparison, we did not perform any (albeit simple) hyperparameter tuning for our method, since we did not perform any hyperparameter tuning for the LieGAN method.
>
> It is also evident that the error bars for the scores in both methods overlap, except for $N=200$ where our method vastly outperforms LieGAN. Thus, the experimental results seem to suggest that our method is comparable to LieGAN in terms of accuracy, except when $N$ is low. Ultimately, the purpose of this experiment was to show that our method competes with SOTA in terms of accuracy when detecting affine symmetries, while offering a computational advantage. However, using the median as the estimate and the IQR for the error bars gives the results in Table 1 of the rebuttal attachment.
>
> We now present an additional experiment that uses both linear terms and a few sinusoidal terms as a pre-determined model. First, we generate 2048 numbers $x_i$ and 2048 numbers $y_j$, each from $U(0,2\pi)$. Next, for each pair $(x_i,y_i)$, we obtain $z_i$ by means of $z_i = \sin(x_i)-\cos(y_i)$, so that a ground-truth level set description of the data is given as $z-\sin(x)+\cos(y) = 0.$
>
> We first apply our level set estimation method to estimate this level set. We optimize the coefficients of the model
>     \begin{equation*}
>         a_0 + a_1 x + a_2 y + a_3 z + a_4 \cos(x) + a_5 \cos(y) + a_6 \cos(z) + a_7 \sin(x) + a_8 \sin(y) + a_9 \sin(z)=0
>     \end{equation*}
>     subject to $\sum_{i=0}^9 a_i^2 = 1$. In light of Eq. (10) in the paper, the matrix $B$ has a row for each of the 2048 tuples $(x_i,y_i,z_i)$, and 10 columns, which columns correspond to the 10 different feature functions in our pre-determined model. The vector $w$ contains all 10 parameters $\\{a_i\\}_{i=0}^{9}$.
>
> Using the $L_1$ loss function and the (Riemannian) Adagrad optimization algorithm with learning rate $0.01$, our estimated level set description is
>
> \begin{equation*}
>         -0.57737 z - 0.57713 \cos(y) + 0.57756\sin(x) = 0,
> \end{equation*}
> which is approximately equivalent to the ground truth answer up to a scaling factor.
>
> **Relevant reference:** We will include a brief discussion of the paper by Desai et al. We note that LieGAN outperformed this work in their experiments, and thus a comparison is unnecessary.
>
> **Details about Section 4.2 experiment**:  For this experiment, the search space for level set estimation was limited to cubic polynomials, while the search space for vector fields was limited to quadratic coefficients. In this setting, the only valid symmetries are constant multiples of the given ground truth symmetry. It is a good point that $fX$ may not have a favorable cosine similarity. Thus, it seems that the cosine similarity metric can only be applied where the issue of uniqueness can be controlled, as in this particular example. We will discuss this limitation in the revision. The purpose of this experiment is to show that our method can detect non-affine symmetries, where current SOTA cannot. See also our overall rebuttal for further discussion.
>
> [R1] A. Pinkus, “Weierstrass and approximation theory,” Journal of Approximation Theory, vol. 107, no. 1, pp. 1–66, 2000.

---

> ### Comment · Reviewer_S4uK · 2024-08-14
>
> I appreciate the response and the additional discussion by the authors.
> Some of my concerns have been resolved, but some others still remain.
>
> Q1. LieGAN comparison for affine symmetry detection
> By the additional experiment in the rebuttal attachment, now the benefit of the proposed method against LieGAN has become clear. While the weakness about initialization must be discussed in the paper, the result is sufficient to support the benefit.
>
>
> Other concerns are related the "predetermined models". Here I would like to reorganize the other concerns as followings by narrowing down the scope from what are general non-affined symmetries to coefficients of predetermined models of vector fields:
>
> Q2. Is finding vector fields enough to say "we solve non-affine symmetry detection problem"?
>
> Q3. Is finding coefficients of predetermined regression models of vector fields enough to say "we solve non-affine symmetry detection problem"?
>
> Q4. Is the coefficients of predetermined models of vector fields a plausible evaluation metric?
>
> The authors address Q2 and Q3 well. While the authors' discussion about these questions including the new experiment with sin, cos model should be added to the manuscript, Q2 and Q3 cannot be a reason to reject this paper.
> Now I agree that This paper has its own benefit which support possibility to be accepted, including SOTA performance for affine symmetry detection and an approach to reduce non-affined symmetry detection to finding a vector field.
>
>
> However, Q4 is a concern in academic perspective that this paper can lead to misunderstanding among future researchers about what non-affine symmetry is and how it should be evaluated.
> One of reason I asked "The case that pre-determined parametric model of vector fields does not cover the GT vector field used for data generation is not discussed." is that I wanted the authors to realize the cosine similarity metric for coefficients do not have any sense for that case and to provide another plausible metric. The universal approximation property does not defense this issue.
>
> Let me explain why the evaluation metric of cosine similarity for coefficients is weird by analogy to another formulation of problem. Suppose that there is a computer vision method where a neural network produces output images. Then the method must be evaluated using an image distance metric between output images from the proposed network and GT images. On the other hand, it will be so weird and misleading if one assume there is an GT network parameters which exactly produces GT images and evaluating the method using distance between network parameters.
>
> It seems illogical to claim the benefit of the non-affine symmetry detection results when there hasn't been sufficient discussion on how non-affine symmetry should be evaluated. This could potentially establish an incorrect evaluation method for future researchers, which might undermine the benefits of the proposed method.
>
> For this reason, my current score is "borderline reject". The minimum requirement for acceptance is as following:
> * Clarify that: the evaluation metric is incomplete since it is not defined on vector fields themselves but depends on choice of parametric representations of vector fields. Then large changes in vector fields might raise relatively small changes in coefficients and vice versa.
> * At least one additional experiment for the case that pre-determined parametric model of vector fields does not cover the GT vector field used for data generation. Then the authors should clarify they currently do not know plausible evaluation metric for this case and finding such metric is an important future work, and qualitative results which shows GT and estimated vector fields must be included.

---

> > ### Author Response · Authors · 2024-08-14
> >
> > We thank the reviewer for their response. First, we acknowledge the reviewer's point that the cosine similarity evaluation metric used in the non-affine symmetry detection experiment is incomplete. When a GT vector field is not strictly expressible in terms of a pre-determined model, the evaluation metric used is not applicable. Moreover, the uniqueness issue with vector fields is not addressed using this metric, since vector fields $X$ and $fX$ are not generally evaluated equally.
> >
> > Second, we provide an experiment in which the GT vector field is not recovered by our method. For this, we use the dataset provided in a previous response, where 2048 numbers $x_i$ and 2048 numbers $y_i$ are generated, each separately from $U(0,2\pi)$. Numbers $f_i$ are calculated by means of $f_i=\sin(x_i)-\cos(y_i)$, and we seek to identify a symmetry of $f(x,y) = \sin(x)-\cos(y)$. A GT vector field which characterizes the symmetry of $f$ is given as
> >
> > \begin{equation*}
> >        X = \sin(y)\partial_x - \cos(x)\partial_y.
> > \end{equation*}
> >
> > Applying our method with a pre-determined model of degree 2 polynomials gives an estimated vector field $\hat{X}$ of
> >
> > \begin{equation*}
> >             \hat{X} = \left( 0.7024-0.1874x-0.2203y+0.0121x^2+0.0242xy+0.0133y^2 \right) \partial_x
> > \end{equation*}
> > \begin{equation*}
> >             + \left( -0.5783+0.2665x+0.1236y-0.0311x^2-0.0150xy-0.0097y^2 \right) \partial_y.
> > \end{equation*}
> >
> > This result was obtained using the L1 Loss function, the Riemannian Adagrad optimizer with learning rate $0.1$, training for $5000$ epochs. It is clear that the estimated vector field does not cover the GT vector field. Moreover, the limitations of the cosine similarity as used in experiment 4.2 are evident, since this evaluation metric cannot be applied in this case.
> >
> > As the reviewer has said, this leaves the problem of finding suitable evaluation metrics as an open problem in symmetry detection, since no suitable evaluation metrics have been applied in a general setting. We will clarify this point in the revision.
> >
> > Third, we offer the following improved evaluation metric as a plausible alternative. A cosine similarity can be defined between two smooth functions directly, without using parameters in a pre-determined model. The set of $\mathcal{C}^{\infty}$ functions on a closed, bounded subset $\Omega$ of $\mathbb{R}^n$ forms a vector space which can be equipped with the inner product defined by the definite integral of the product of two functions:
> >
> > \begin{equation*}
> >             \langle f,g \rangle = \int_{\Omega} fg dx.
> > \end{equation*}
> >
> > Thus, with a norm induced by this inner product, a cosine similarity between functions $f$ and $g$ can be obtained by means of
> >
> > \begin{equation*}
> >             \cos(\theta) = \dfrac{\langle f,g \rangle}{||f|| \cdot ||g||}.
> > \end{equation*}
> >
> > An improved evaluation metric for vector fields is thus an aggregation (such as the mean) of cosine similarity scores for each component pair of the vector fields. Concretely, given vector fields $X=f_i\partial_{x^i}$ and $Y=g_i\partial_{x^i}$, their similarity can be estimated by
> >
> > \begin{equation*}
> >             \text{sim}\left( X,Y \right) = \dfrac{1}{N}\sum_{i=1}^{N} \dfrac{| \langle f_i,g_i \rangle |}{||f_i|| \cdot ||g_i||},
> > \end{equation*}
> > which will take values in $[0,1]$, with $0$ signifying minimal similarity and $1$ signifying maximal similarity.
> >
> > In our example, $x_i,y_i \in [0,2\pi]$, so that a suitable domain for integration is $[0,2\pi] \times [0,2\pi]$. (The distribution of the data defines this domain of integration in general.) We apply the above formula using the GT vector field and our estimated vector field and obtain a similarity score of approximately $0.62$. For the results of the final trial in experiment 4.2, the similarity scores would be adjusted from $0.9919$ to $0.9976$ (ours) and $0.4930$ to $0.4583$ (LieGAN's).
> >
> > Our new similarity score is imperfect, since the multiplication of $X$ by any smooth function is a valid GT vector field. However, this new similarity score is an improvement over the original, since it does not rely on pre-determined models. We offer this improved similarity score as a plausible alternative to the parameter-based cosine similarity score used in experiment 4.2.
> >
> > Another possible evaluation metric would make use of the point-wise inner product of two vector fields, as suggested in the reviewer's previous comment about an average of cosine similarities for vectors at each point. This may be suitable, though the inner product of two tangent vectors can only be computed if a metric tensor is given: thus, this method would need to make an assumption about the metric tensor for the dataset. As we discuss in appendix A, it is common in machine learning to assume that the metric tensor is the Euclidean metric. However, we believe, as we mention in the appendix, that this commonly-accepted assumption may eventually be challenged, and so we prefer the first alternative evaluation metric.

---

> ### Comment · Reviewer_S4uK · 2024-08-14
>
> I appreciate for the authors' anthusiastic discussion and additional experiment. Clarifying my conclusion first, I raised my score to weak accept.
>
> The first version of the manuscript evaluated symmetry detection in the coefficient space of a predetermined model of vector fields, but the authors finally proposed the way to evaluate in the space of vector fields. The reason why I leaned toward rejection was based on the coefficient space evaluation. The latter evaluation is much technically sound than the former one, for me.
> Of course evaluating on the space of vector fields still has limitations, since handling symmetries in the sense of {$S$ | $f\circ S = f$} and the sense of the vector fields are not equivalent e.g. X and fX.
> However, overcoming these remained challenges is considered as a good future work, but the authors are not needed to provide a method and results to overcome them. I think attaching relavent discussions from this rebuttal & discussion period into the revision will be fine.
>
> There are also remained concerns which I hope the authors to adress them in the revision (not need to response during this discussion period, which has only one hour left).
> * The vector field similarity depends on the choice of the bounding volume. It will be clarified.
> * For $\mathrm{sim}\left(X,Y\right)$, The current formula seems to depend on choice of coordinates so that lacking geometric meaning. Please check. I think the summation symbol should appearr three times, once in the numerator and twice in the denominator.

---

### Author Rebuttal · Authors · 2024-08-07

We thank all of the reviewers for their very helpful comments. We first wish to emphasize here our novel contribution to symmetry detection, particularly in light of current SOTA methods. To date, the most successful method of detecting continuous symmetry is LieGAN, which is a large neural network that can successfully detect affine symmetries in low dimensions. In contrast, our method uses polynomial regression, with constrained coefficients, to detect not only affine symmetries, but symmetries of much higher complexity. The computational advantage alone, owing to our novel use of vector fields, offers a highly non-trivial advantage to current SOTA methods. Moreover, we appear to be the first to detect any continuous symmetries beyond affine transformations using only training data. The current SOTA method (LieGAN), after decades of work on the subject, uses a GAN to detect rotational symmetry in two dimensions, where such a transformation is likely the most rudimentary of all symmetries, from a mathematical perspective.

Our experiments use polynomial regression, though we note that our method can be trivially adapted to accommodate different functions. We use polynomials not only for better comparison with current SOTA, (which, if presented in terms of vector fields, would assume linear polynomial vector field components) but also due to the universal approximation property of polynomials, as polynomials can approximate any smooth function to an arbitrary degree of accuracy [R1]. We have also included an experiment which uses a non-polynomial basis in response to a specific reviewer's concern for this.

We also wish to address a technical question relating to the potential need for careful handling of non-affine symmetry detection. Given a vector field $X$ and a vector field $fX$, a function $h$ is $X$-invariant if and only if it is $fX$-invariant. If $h$ is $X$-invariant, the flow of $X$ is a symmetry of $h$, so that the flow of $fX$ is also a symmetry of $h$: this follows from our discussion of vector fields.

For example, a circle centered at the origin, characterized by $F=0$ where $F(x,y) = x^2+y^2-r^2$ for some real number $r$, exhibits rotational symmetry described by $X=-y\partial_x + x\partial_y$. However $\frac{1}{y} X (F) = 0$, so that the flow of the vector field $-\partial_x + \frac{x}{y} \partial_y$, where defined, is also a symmetry of the circle.

This may seem to introduce a theoretical problem requiring greater care when detecting non-affine symmetry. However, our proposed method of constructing models which are invariant with respect to the symmetry group requires only the identification of functions which are invariant with respect to the vector fields. Thus, both $X$ and $fX$ are valid answers for "ground truth" symmetries, since a function $h$ is $X$-invariant if and only if it is $fX$-invariant.

This lack of uniqueness does present a challenge when estimating suitable vector fields. When estimating the symmetries via constrained regression, $X$ and $fX$ may or may not both be present in the search space. As we allude to on page 5, symmetry estimation can be done symbolically, and this would eliminate the possibility of both $X$ and $fX$ appearing in the set of discovered vector fields. A non-symbolic technique is also discussed, with the challenge being partially addressed in lines 226-231, though this issue may also warrant a discussion in a new ``limitations'' subsection of the methods section.

We also briefly mention that this issue can be addressed by identifying "special" vector fields and reducing the search space of vector fields to linear combinations of these special vector fields known as Killing vectors. This would eliminate the uniqueness problem and is experimented with in Appendix A. This approach is a special case of symmetry detection where only isometries can be detected.

Another point which may help to address concerns about our handling of the difficult nature of non-affine symmetry detection is in the relationship between the flows of $X$ and $fX$ generally. The trace of the flow of $X$ through the point $p$ is characterized by the level set $h_i=c_i$, where $\{h_i\}$ is a complete set of $X$-invariant functions. Since a function is $X$-invariant if and only if it is $fX$-invariant, a complete set of invariant functions for $fX$ can be taken to be $\{h_i\}$ without loss of generality. Thus, the level set $h_i=c_i$ also characterizes the trace of the flow of $fX$ through the point $p$. This argument assumes that the flows of $X$ and $fX$, as well as the vector fields themselves, are well-defined in an open neighborhood about the point $p$.

In fact, in our example, the flow of $-\partial_x + \frac{x}{y} \partial_y$, assuming $y>0$, is given as $\Phi(t,(x,y)) = \left(x-t, \sqrt{y^2+2tx-t^2} \right)$, which trace is a (part of a) circle (we note that the flow parameter is $-x$). The trace of this flow through a point is equivalent to the trace of $X$ through the same point.


[R1] A. Pinkus, “Weierstrass and approximation theory,” Journal of Approximation Theory, vol. 107, no. 1, pp. 1–66, 2000.

---

### Comment · Area_Chair_sZCb · 2024-08-10

Dear reviewers,

Could you please respond to the rebuttal, discuss with the authors and finalize your score?

---

### Comment · Area_Chair_sZCb · 2024-08-13

For those who have not responded to the rebuttal, could you finalize your score and response by Aug 13 (tomorrow)? Thanks!

your AC

---

### Decision · Program_Chairs · 2024-09-25

**Decision:**

Accept (poster)

**Comment:**

This paper introduces a novel method for detecting symmetries in data using vector fields, extending beyond traditional affine transformations. The approach is innovative and addresses an important problem, with potential for significant impact. While one reviewer raised concerns about the clarity of the presentation and reliance on toy datasets, other reviewers acknowledged the method's strengths, particularly its effectiveness in small-sample scenarios and its computational efficiency compared to existing methods like LieGAN. Despite the presentation issues, the paper's contribution is valuable, and I recommend accepting it.